# WHY CAN'T TRANSFORMERS LEARN MULTIPLICATION? REVERSE-ENGINEERING REVEALS LONG-RANGE DEPENDENCY PITFALLS

## ABSTRACT

Language models are increasingly capable, yet still struggle at a seemingly simple task of multi-digit multiplication. In this work, we study why, by reverse-engineering models that successfully learns multiplication via either *implicit chain-of-thought* or *"1-to-N"* digit multiplication, and report three findings: (1) Evidence of long-range structure: Logit attributions and linear probes indicate that the model encodes the necessary long-range dependencies for multi-digit multiplication. (2) Mechanism: the model encodes long-range dependencies using attention to construct a directed acyclic graph to "cache" and "retrieve" pairwise partial products. (3) Geometry: the model implements partial products in attention heads by forming Minkowski sums between pairs of digits, and digits are represented using a Fourier basis, both of which are intuitive and efficient representations that the standard training model lacks. With these insights, we revisit the learning dynamics of standard training and find that the model converges to a local optimum that lacks the required long-range dependencies. We further validate this understanding by introducing an auxiliary loss that predicts the "running sum" via a linear regression probe, which provides an inductive bias to learn long-range dependencies and enables the model to successfully learn multi-digit multiplication. In summary, by reverse-engineering the mechanisms of both an implicit chain-of-thought model and a "1-to-N" multiplication model, we uncover a pitfall for learning long-range dependencies in Transformers and provide an example of how the correct inductive bias can address this issue.

## 1 INTRODUCTION

Large language models demonstrate striking capabilities across reasoning, planning, and tool use. Yet, they also struggle on surprisingly simple algorithmic tasks (Nye et al., 2021; Lee et al., 2023). Why do Transformers excel at some tasks, but struggle with others? One such example is multi-digit multiplication. Despite having *billions* of parameters, models like Llama-3.2 90B or GPT4 still fail at 4x4-digit multiplication (Gambardella et al., 2024),[1] even when explicitly fine-tuned on the task (Yang et al., 2023). Why do Transformers struggle with multi-digit multiplication?

We study these questions by contrasting a standard trained model (ST), which fails at multiplication, with two models that succeed. In particular, researchers have shown various "tricks" to get Transformers to perform multiplication. One example is to train with *implicit chain-of-thought* (ICoT) (Deng et al., 2024; 2023), which works by providing explicit chain-of-thought tokens during training, but gradually remove them and thus force the model to internalize intermediate steps in its latent states. A second approach is to provide examples of shorter digits in the training data: as opposed to just presenting NxN-digit multiplication, one can include examples of shorter digits: 1x1, ..., 1xN, 2x2 ... 2xN, and so on(Shen et al., 2023). We refer to this approach as *1-to-N* digit multiplication. Note that both of these methods can be viewed as deconstructing the task into simpler sub-tasks.

We reverse-engineer the two successful models, the ICoT and 1-to-N models, and uncover several insights. First, unlike the ST model, the successful models learn the correct long-range structure

---

Code: `https://anonymous.4open.science/r/icot-F822`
[1] Note that some recent proprietary models that do solve multi-digit multiplication may rely on tool-use.

needed for multi-digit multiplication. We provide evidence of this using logit attributions and linear regression probes. *Mechanistically*, successful models encode long-range dependencies by organizing its attention into a sparse, binary-tree-like graph, which (i) selects the correct digit pairs to compute partial products and (ii) "caches" these intermediate computations into earlier tokens for later retrieval. Lastly, *geometrically*, attention heads realize the sub-tasks (partial products) as Minkowski sums of digit embeddings, and represent digits with Fourier bases, yielding a pentagonal prism structure – both of which are intuitive and efficient representations that the ST model lacks.

With these insights, we revisit the dynamics of standard training: under gradient descent and an auto-regressive loss, the model never learns these long-range dependencies, and thus loss plateaus on the middle digits. To confirm our understanding, we introduce a simple fix by introducing an auxiliary loss that supervises the model to predict a "running partial sum" through a lightweight linear regression probe. This can be viewed as not only an alternative way of deconstructing the task into sub-tasks, but also as an inductive bias to learn the proper long-range dependencies, allowing the model to achieve perfect accuracy.

In summary, by reverse-engineering networks that successfully implement multi-digit multiplication, we uncover how it implements long-range dependencies, a mechanism that the unsuccessful model lacks. Our work highlights a challenge for Transformers to learn long-range dependency using gradient descent and an auto-regressive loss. While we demonstrate a task-specific inductive bias to address this issue, we anticipate generic improvements to address this limitation.

## 2 EXPERIMENT SETUP, TRAINING PROCEDURES, NOTATIONS

**Task, Models.** We are interested in understanding the differences in models trained with standard training, ICoT, and 1-to-N training. Standard training (ST) refers to training a model on samples of NxN multi-digit multiplication, with a standard auto-regressive loss. ICoT provides intermediate chain-of-thought (CoT) tokens as part of the training data, but as training progresses, the CoT tokens are gradually removed. 1-to-N training refers to training a model on samples of not only NxN digit multiplication, but also on samples of shorter digits, i.e., 1x1 ... 1xN, 2x1 ... 2xN, etc. From experiments, we find that the simplest multi-digit multiplication in which standard training fails but ICoT and 1-to-N training works is $4{\times}4$ digit multiplications. Similarly, the smallest architecture in which ICoT and 1-to-N training works is a 2-layer model with 4 attention heads. All of our models are trained from scratch. Thus we carefully study 2-layer 4-head models trained with standard training, ICoT, or 1-to-N training on $4{\times}4$ multiplication.

**Training Procedures: ICoT** Our ICoT setup is the same as that of Deng et al. (2024). Here we provide an informal overview of ICoT, with details in Appendix A.1. Assume two operands $a = (a_3, a_2, a_1, a_0)$, $b = (b_3, b_2, b_1, b_0)$ and their product $c = (c7 \dots c_0)$. Operands are written least-significant digit first, similar to other algorithmic setups (Deng et al., 2024; 2023; Lee et al., 2023).

For ICoT, the training data includes intermediate chain-of-thought (CoT) tokens $q_i$ that explicitly record the step-by-step calculations. As a simple illustration, consider $12 \times 34$. The tokens appearing between the two equal signs follow the same CoT format used in our $4 \times 4$-digit multiplication tasks:

$$12 * 34 = \underbrace{48}_{12*4} + \underbrace{360}_{12*30} \underbrace{(408)}_{\text{running sum}} = 408$$

At each training epoch, a fixed number of CoT tokens are removed from the *left* of the chain. Concretely, the training examples at each epoch may have the following form:

$$
\begin{aligned}
&\text{(Epoch 1)} \quad a_0a_1a_2a_3 * b_0b_1b_2b_3\%\%\% \; q_0 \dots q_i \dots q_j \dots q_k \dots q_\tau \;\#\#\#\# \; c_0 \dots c_7 \\
&\text{(Epoch 2)} \quad a_0a_1a_2a_3 * b_0b_1b_2b_3\%\%\% \; q_i \dots q_j \dots q_k \dots q_\tau \;\#\#\#\# \; c_0 \dots c_7 \\
&\text{(Epoch 3)} \quad a_0a_1a_2a_3 * b_0b_1b_2b_3\%\%\% \; q_j \dots q_k \dots q_\tau \;\#\#\#\# \; c_0 \dots c_7 \\
&\qquad \cdots \\
&\text{(Epoch N)} \quad a_0a_1a_2a_3 * b_0b_1b_2b_3\%\%\% \;\#\#\#\# \; c_0 \dots c_7
\end{aligned}
$$

where $q_i$ are CoT tokens and $\%, \#$ are special delimiters.[2] Note that after each epoch, the model sees a shorter chain by truncating some tokens, and that by the end, only the operands and final

---

[2] These delimiters have no special meaning beyond matching the setup of Deng et al. (2024).

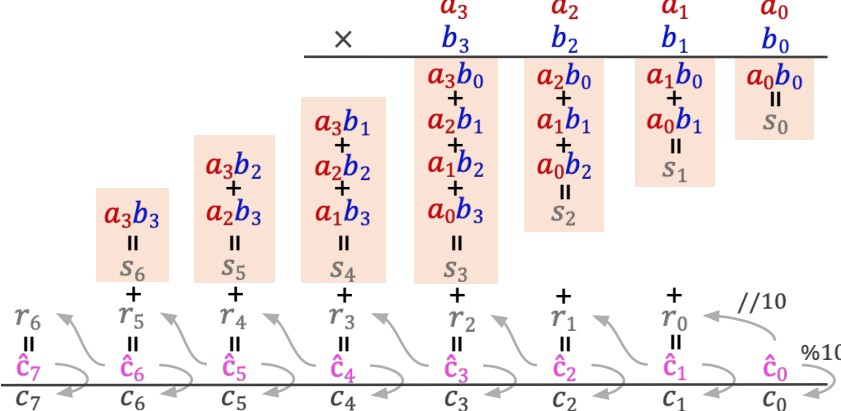

Figure 1: **Multiplication has long-range dependencies**, which can be captured by an intermediate value $\hat{c}_i$, from which both the solution ($c_i$) and carries ($r_i$) can be derived from.

answer remain. For comparison, standard fine-tuning only trains on the operands: $a_0 a_1 a_2 a_3 *$ $b_0 b_1 b_2 b_3 \%\%\%\#\#\#\# c_0 \ldots c_7$.

Interestingly, the ICoT model is able to achieve 100% accuracy on 4×4 digit multiplication, while standard fine-tuning only achieves less than 1% accuracy. Note that scaling does not help – scaling to a 12 layer 8 head model achieves the same $< 1\%$ accuracy, and Yang et al. (2023) show that fine-tuning a 2B model still plateaus at 95% accuracy.

For more details regarding training (data format, sample size, hyperparameters), see Appendix A.

**Training Procedures: 1-to-N.** An alternative approach to teach Transformers is to train on "1-to-N" digits of multiplication (Shen et al., 2023): rather than training only on samples of 4x4 digit multiplication, we also train on shorter digits, i.e., 1x1, ..., 1x4, 2x1, ..., 2x4, etc. Similar to ICoT, digits are written least-significant digit first. Other than including simpler examples, all other steps are the same as standard training.

**Notations.** We use plain letters $(a, b)$ for scalars, bold lowercase letters $(\mathbf{x}, \mathbf{h})$ for vectors, bold uppercase letters for matrices $(\mathbf{X})$, and calligraphics $(\mathcal{A}, \mathcal{B})$ for sets. $\mathbf{h}_t^\ell$ indicates the hidden states at layer $\ell$ timestep $t$. Timesteps for solution tokens $c_k, k = [0, \ldots, 7]$ are notated $t_{c_k}$. $\text{ATT}_h^\ell(\cdot)$, $\text{MLP}^\ell(\cdot)$ indicate the output of the attention heads or MLP blocks at layer $\ell$, head index $h$. $\mathbf{E} \in \mathbb{R}^{V \times d}$ indicate digit embedding weights.

## 3 COMPARING THE MECHANISMS OF ICoT, 1-TO-N, AND ST

### 3.1 LONG-RANGE DEPENDENCIES IN MULTI-DIGIT MULTIPLICATION

Here we discuss how one might solve multi-digit multiplication, and the required long-range dependencies needed to solve multiplication.

One approach to compute each digit, $c_k$, is as follows:

$$s_k \triangleq \underbrace{\sum_{i+j=k} a_i b_j}_{\text{sum of partial products}}, \qquad c_k = (s_k + r_{k-1}) \bmod 10, \qquad r_k = \underbrace{\left\lfloor \frac{s_k + r_{k-1}}{10} \right\rfloor}_{\text{carry}}, \qquad r_{-1} = 0 \qquad (1)$$

Note that both $c_k$ and $r_k$ can be expressed with an intermediary term $\hat{c}_k$, which encapsulates both the relevant information from the partial products and the carry:

$$\hat{c}_k \triangleq s_k + r_{k-1}, \qquad c_k = \hat{c}_k \pmod{10}, \qquad r_k = \left\lfloor \frac{\hat{c}_k}{10} \right\rfloor \qquad (2)$$

Importantly, note the *long-range dependencies* needed for multi-digit multiplication. Specifically, we highlight two observations: **(i)** To determine $c_k$, one must use all the partial products $\{a_i b_j | i +$

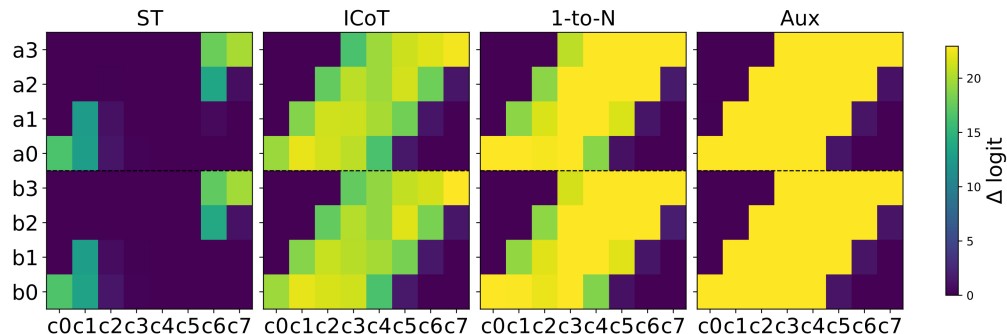

Figure 2: **Logit Attribution.** We test for whether each model has correctly learned long-range dependencies by measuring how sensitive the logits of output digits $c_i$ are to each operand digit (i.e., $a_i, b_j$). This is done by measuring the change in $c_i$'s logits when a single operand digit is perturbed.

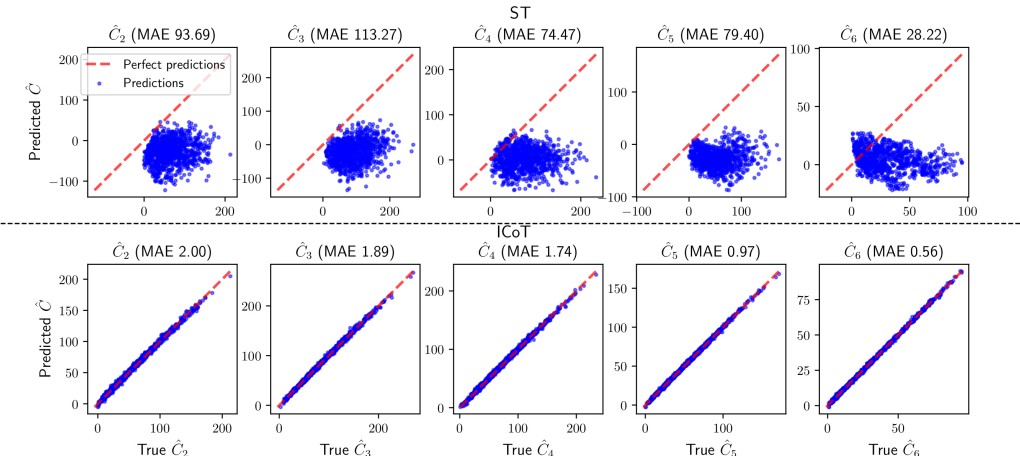

Figure 3: **Linear regression probing results for $\hat{c}$.** We probe from the middle of the last Transformer block, after attention heads but before MLPs.

$j \leq k\}$, since all of these terms contribute to $c_k$. **(ii)** Knowing the intermediary term $\hat{c}_k$ suffices to compute $c_k$ and to propagate necessary information for later digits. Thus we use $\hat{c}_k$ as a probing signature (Section 3.2) at each timestep $t_{c_k}$ to check if the model is utilizing all the necessary long-range information to predict the correct tokens $c_k$.

In the following sections, we demonstrate how the ICoT and 1-to-N models satisfy such long-range dependency while the standard training model does not.

## 3.2 Evidence of Long-Range Dependencies in Successful Models

We first demonstrate two lines of evidence that the ICoT and 1-to-N models satisfy long-range dependencies in multi-digit multiplication, while the standard training model does not.

**Logit Attributions.** Note from Figure 1 that digits $a_i, b_i$ can only affect $c_k$ terms where $k \geq i$. Also note that at timestep $t_{c_k}$, the pairwise products $\{a_i b_j | i + j = k\}$ affect the final prediction $c_k$ the most. "Earlier" pairwise products $\{a_i b_j | i + j \leq k\}$ can still affect $c_k$, but with diminishing effects as $i + j$ gets smaller.

We directly test for these relationships in our ICoT, 1-to-N, and ST models using logit attributions. Namely, given an input sample ORIG $:= a_0 a_1 a_2 a_3 * b_0 b_1 b_2 b_3$, we measure the logits of the model's predictions for $c_{0-7}$ : $\mathrm{logit}_{c_k}(\mathrm{ORIG})$. We then randomly swap out one of the operand digits at timestep $t$ (e.g., $\tilde{a}_2$) to construct a counterfactual input COUNTER$_t = a_0 a_1 \tilde{a}_2 a_3 * b_0 b_1 b_2 b_3$ and measure the change in logits: $\Delta_{t,k} = \mathrm{logit}_{c_k}(\mathrm{ORIG}) - \mathrm{logit}_{c_k}(\mathrm{COUNTER}_t)$ Thus $\Delta_{t,k}$ measures the effect that digit at timestep $t$ has on the prediction of token $c_k$.

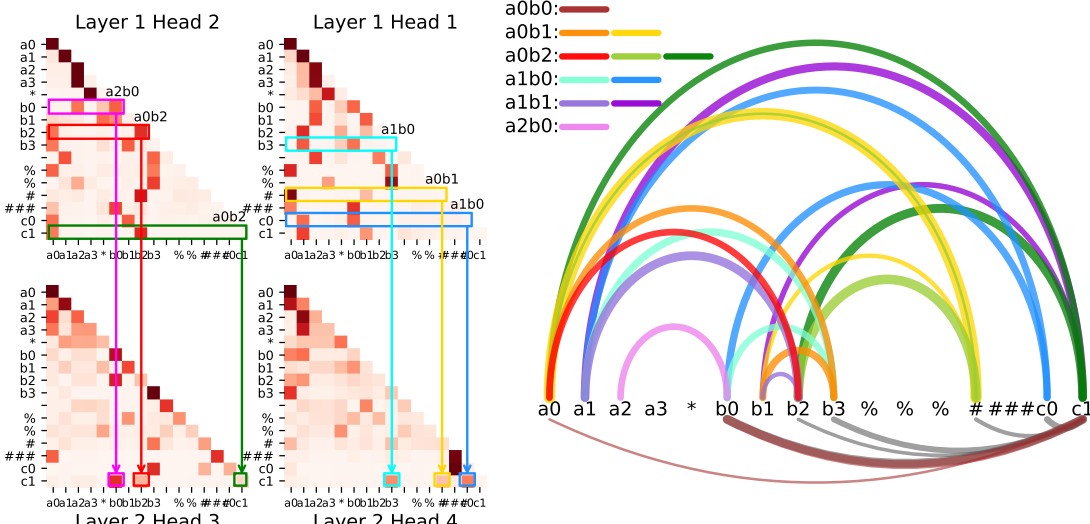

Figure 4: **Visualization of attention tree to compute $c_2$.** Left: Attention maps for selected heads show the first layer "cache" pairwise products ($a_i b_j$) across earlier timesteps, from which the second layer reads from (Not all tree paths are shown). Right: A visualization of the attention tree. Each arc indicates tokens being attended to at specific timesteps. Colored arcs above and below the digits indicate attention patterns from the first and second layers respectively. Example: orange arc indicates that at timestep $b_3$, the model attends to $a_0$ and $b_1$, from which the second layer reads from.

We use 1,000 samples for each $(t, k)$ pair and show the results in Figure 2. For ST, the model does not see the correct dependencies between earlier tokens to middle tokens, while the ICoT and 1-to-N models do, suggesting that the model has indeed learned the correct long-range dependencies.

**Probing for $\hat{c}_k$.** Note from Figure 1 and Equation 2 that the long-range dependencies can be captured by an intermediate term, $\hat{c}_k$. We test for whether $\hat{c}_k$ information can be decoded from the hidden states of the models using linear regression probes. Namely, at each timestep $t_{c_k}$ we predict for $\hat{c}_k$ by training a single vector $\mathbf{w}_k \in \mathbb{R}^d$ such that $\mathbf{w}_k \mathbf{h}^{2.\text{mid}}_{t_{c_k}} = \hat{c}_k$ using a MSE loss, where $\mathbf{h}^{2.\text{mid}}$ is the hidden state at layer 2 after attention heads, before MLPs.

Figure 3 reports the mean absolute error from probing for $\hat{c}_k$ for middle and late digits, $k = 2, \ldots, 6$, for the ST and ICoT model (see Figure 9 for 1-to-N results). Note that the accuracy from the ICoT model is much higher than that of ST, further suggesting that the ICoT model has learned the correct long-range dependencies while ST has not.

### 3.3 ENCODING LONG-RANGE DEPENDENCIES VIA ATTENTION TREES

How do the successful models compute long-range dependencies? Here we describe how the model's attention patterns induce a shallow directed acyclic graph, akin to a binary expression tree, in order to encode long-range dependencies.

Namely, in the first layer, across all timesteps $t > 5$,[3] each attention head only attends to a *pair* of digit tokens, $\{a_i, b_j\}$ (Figure 4, left). This allows the model to produce the pairwise product $a_i b_j$ (see Section 4.1 for *how* attention heads represent pairwise products), but also allows the model to *cache* the product $a_i b_j$ in the hidden state of layer 1 at timestep $t$ (i.e., $\mathbf{h}^1_t$). Put differently, product pairs $\{a_i b_j\}_{i,j \in \{0,\ldots 3\}}$ are "cached" in the first layer across different timesteps ($\mathbf{h}^1_t, t < t_{c_k}$).

At later timesteps $t \geq t_{c_k}$, when the model predicts solution tokens $c_k$, this allows the second layer attention heads to attend to a small set of previous *cache sites*, i.e., where the appropriate pairs of products $a_i b_j, i + j = k$ are stored from earlier timesteps.

**Example:** Figure 4 depicts the attention patterns when the model predicts $c_2$, given input "$a_{0\ldots3} * b_{0\ldots3} = c_0 c_1$". These attention maps are averaged from 1,000 samples from a held out test set. The necessary terms to compute $c_2$ are $a_2 b_0, a_1 b_1, a_0 b_2$, and $\hat{c}_1$ (which in turn requires $a_1 b_0, a_0 b_1, a_0 b_0$).

---

[3]Note that only after timestep 5, both $a$ and $b$ tokens appear in the context.

We use $\text{ATT}_h^{\ell}$ to notate attention head at layer $\ell$, head index $h$. Attention heads $\text{ATT}_3^2$, $\text{ATT}_4^2$ each attend to positions $(b_0, b_2, c_1)$ and $(b_3, \text{``\#''}, c_0)$. Inspecting what was "cached" in the first layer at those timesteps reveals the necessary partial products to compute $c_2$. For example, at timestep $b_0$, $\text{ATT}_1^1$, $\text{ATT}_2^1$ attend to $a_2, b_0$; at timestep $b_2$ $\text{ATT}_1^1$ attends to $a_1, b_1$ while $\text{ATT}_2^1$ attends to $a_0, b_2$; at timestep $c_0$ $\text{ATT}_1^1$ attends to $a_1, b_0$, $\text{ATT}_2^1$ attends to $a_0 b_1$. Thus the model can derive partial products, $a_2 b_0, a_1 b_1, a_0 b_2, a_1 b_0, a_0 b_1$ with its attention tree.[4]

While Figure 4 shows an example of the "attention tree" for predicting $c_2$, one can similarly reconstruct the correct trees for all digits $c_0, \ldots, c_7$ using the attention patterns for all digits in Figure 15. Similarly, the 1-to-N model encodes all the required pairwise products (see Figure 16).

In summary, for each output step $c_k$, the ICoT and 1-to-N models construct a binary-tree-like graph, spread out across timesteps, to attend to the correct pairs of tokens and compute partial products.

## 4 FEATURE GEOMETRY OF ICOT AND 1-TO-N MODELS

In addition to the mechanisms in Section 3, we also study the *geometry* of features in our models. We use examples from our ICoT model in the main text; results for the 1-to-N model can be found in the Appendix (Figures 11, 13).

### 4.1 DIGIT-WISE MULTIPLICATIONS AS MINKOWSKI SUMS

Note from Section 3.3 that the attention patterns are sparse, often only attending to the two positions $a_i, b_j$ being multiplied. In such a case, the outputs of the attention head form a Minkowski sum.

We use plain letters $(x, y, \alpha)$ for scalars, bold lowercase ($\mathbf{a}_x, \mathbf{b}_y \in \mathbb{R}^d$) for vectors, bold uppercase ($\mathbf{W}$) for matrices, and calligraphics $\mathcal{A}, \mathcal{B}$ for sets. $a_{0,\ldots,3}, b_{0,\ldots,3}$ refer to the positions of digits in the two operands. Let $x, y \in \mathcal{N}, \mathcal{N} := \{0, ..., 9\}$ be the actual digit values at positions $a_i, b_j$.

Consider a single head $\text{ATT}$ at the first layer that attends to two positions $a_i, b_j$. We notate its output as $\text{ATT}(i, j)$. Let $\mathbf{W_O} \in \mathbb{R}^{d \times d_{head}}, \mathbf{W_V} \in \mathbb{R}^{d_{head} \times d}$ be the output and value weights of the attention head, $\mathbf{E}[x] \in \mathbb{R}^d$ the token embedding for token $x$, and $\mathbf{a}_x := \mathbf{W_O W_V E}[x], \mathbf{b}_y := \mathbf{W_O W_V E}[y], \mathbf{a}_x, \mathbf{b}_y \in \mathbb{R}^d$.

Now consider the set of possible values for $\mathbf{a}_x, \mathbf{b}_y$ for all tokens:

$$\mathcal{A} := \{\mathbf{a}_x \in \mathbb{R}^d : x \in \{0, ..., 9\}\}, \quad \mathcal{B} := \{\mathbf{b}_y \in \mathbb{R}^d : y \in \{0, ..., 9\}\}, \tag{3}$$

Because the attention head sparsely only attends to two positions, $a_i, b_j$ by $\alpha\%$ and $(1 - \alpha)\%$ respectively, the set of all outputs from the head can be expressed as

$$\mathcal{Z} := \{\mathbf{z}_{xy} : x, y \in \{0, ..., 9\}\}, \tag{4}$$

$$\mathbf{z}_{xy} := \text{ATT}(i, j) = \alpha \mathbf{a}_x + (1 - \alpha)\mathbf{b}_y + \epsilon_{xy}, \quad \mathbf{z}_{xy} \in \mathbb{R}^d. \tag{5}$$

Let us define a scalar multiplication on a set of vectors as $\lambda \mathcal{A} := \{\lambda \mathbf{a} : \mathbf{a} \in \mathcal{A}\}$, and denote the Minkowski sum between two sets of vectors as $\mathcal{S} \oplus \mathcal{T} := \{\mathbf{s} + \mathbf{t} : \mathbf{s} \in \mathcal{S}, \mathbf{t} \in \mathcal{T}\}$. Ignoring position embeddings, the output of the attention head can then be expressed as a Minkowski sum:

$$\text{ATT}(i, j) \subseteq (\alpha \mathcal{A}) \oplus ((1 - \alpha)\mathcal{B}) \oplus \mathcal{E} \tag{6}$$

where $\mathcal{E} := \{\epsilon_{xy}\}$. See Figure 5 (a) for a visualization.

Visually, 3D PCAs can reveal nested representations. Namely, we can observe clusters, each cluster corresponding to a feature (i.e., $a_i$). These clusters form a "global" geometry. When zoomed in to each cluster, we observe additional clusters for a second feature (i.e., $b_j$) that form a "local" geometry of the same shape as its global counterpart. See Figure 5 (b-d) for examples. We provide an explanation for this phenomenon in Appendix D.

---

[4]Note that there may be a couple of different ways that $a_0 b_0$ is derived. One possibility is to re-use $a_0, b_0$ information that was fetched at various timesteps. Another possibility is when $a_0$ is slightly attended to at $\text{ATT}_3^2$ (difficult to see in our visuals). Note that $a_0 b_0$ plays a relatively minor role in computing $c_2$ compared to all other partial products.

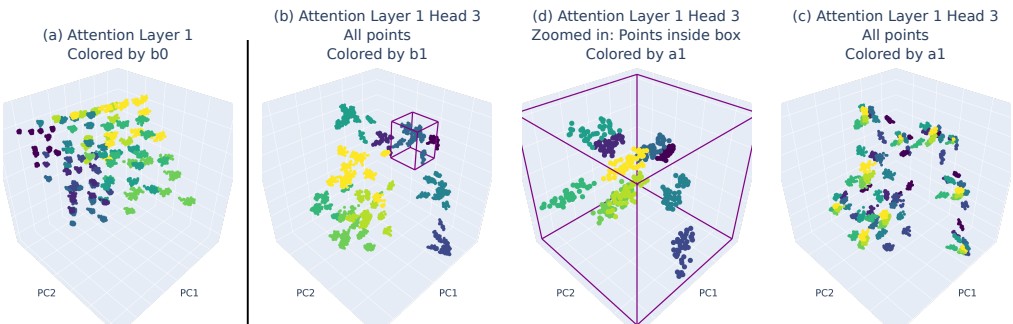

Figure 5: **3D PCA of attention head outputs can form Minkowski sums**, which in turn can form nested representations. Each color represents a different digit. These examples are from the ICoT model. For examples from the auxiliary loss or 1-to-N models, see Figures 10, 11.

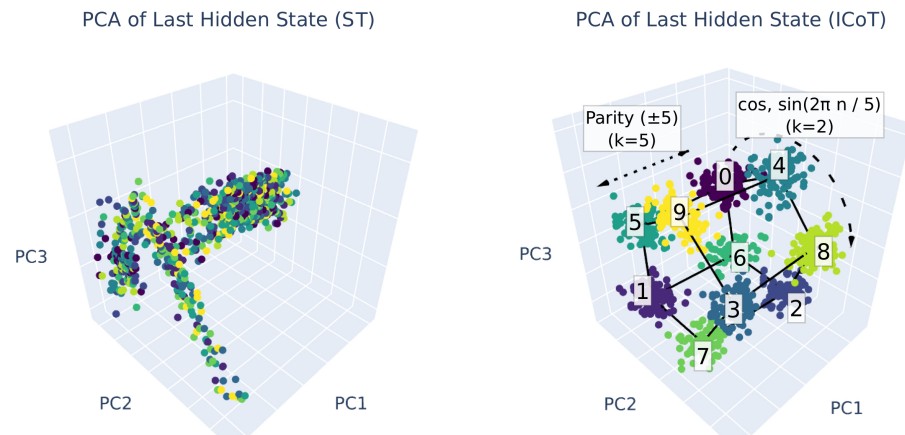

Figure 6: **Digits embedded in a pentagonal prism, using Fourier bases.** No obvious patterns in the SFT model, but the ICoT model encodes digits in a pentagonal prism using Fourier bases.

### 4.2 EMBEDDING DIGITS ON A PENTAGONAL PRISM VIA FOURIER BASES

Similar to Kantamneni & Tegmark (2025), we find that our model encodes digits in Fourier space. Specifically, the model's embeddings $E$, the final hidden layer $\mathbf{h}^L$, and even the weights of the last MLP can be well reconstructed from a small set of Fourier basis functions.

Figure 6 shows a 3D PCA visualization of the final hidden layer at timestep $t_{c_2}$, for both the SFT and ICoT models. While the SFT hidden states do not reveal any obvious patterns, the ICoT hidden states reveal a striking pattern: the ten digits form vertices of a *pentagonal prism*.

This structure is naturally explained by Fourier modes. Consider the Fourier expansion

$$\sum_{k=0}^{9} C_k e^{-2\pi i \frac{kn}{10}}, \quad n = 0, \dots, 9.$$

where $C_n (\neq c_k)$ is some constant per digit $n$. Following Kantamneni & Tegmark (2025), we take frequencies $k \in \{0, 1, 2, 5\}$, yielding the real Fourier basis

$$\Phi(n) = \left[ \underset{(k=0)}{\mathbf{1}(n)} \quad \underset{(k=1)}{\cos\left(2\pi \frac{n}{10}\right)} \quad \underset{(k=1)}{\sin\left(2\pi \frac{n}{10}\right)} \quad \underset{(k=2)}{\cos\left(2\pi \frac{n}{5}\right)} \quad \underset{(k=2)}{\sin\left(2\pi \frac{n}{5}\right)} \quad \underset{(k=5)}{\boldsymbol{p}(n)} \right],$$

where $\mathbf{1}(n) \equiv 1$ is a constant term that always returns 1 regardless of the value of $n$ (i.e., the DC component) and $\boldsymbol{p}(n) \equiv (-1)^n$ is the Nyquist/parity term that alternates between $+1$ and $-1$ across

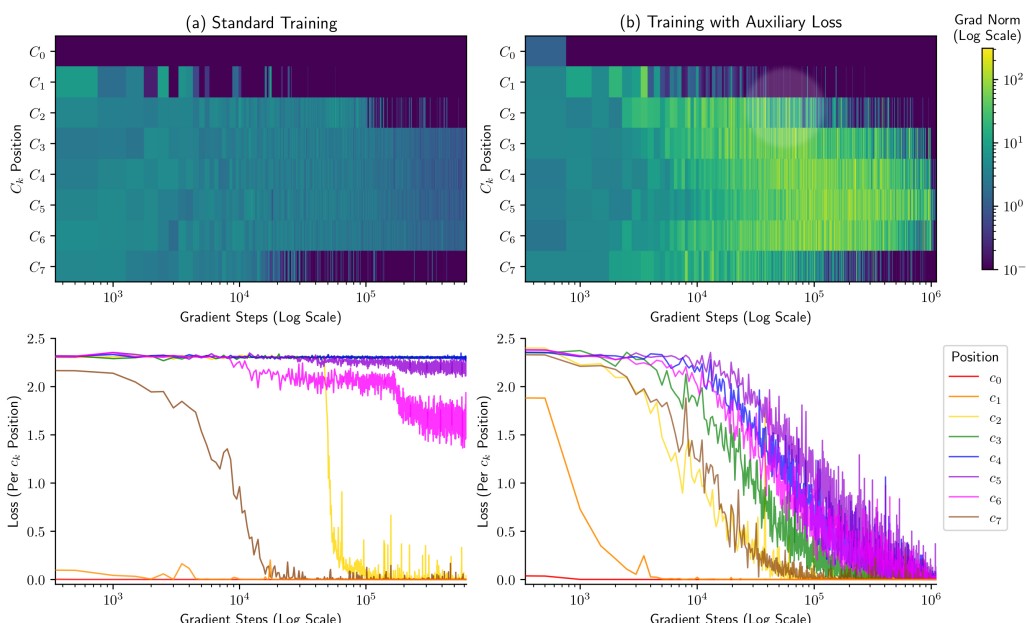

Figure 7: **Gradient norms and losses *per token* $c_k$.** While both methods learn digits $c_0, c_1, c_7$ first, standard fine-tuning gets stuck in a local optimum without having learned the right long-range dependencies, while training with the auxiliary loss allows the model to learn the middle digits.

digits depending on the value of $n$ (i.e., returns $+1$ when $n$ is even, $-1$ when $n$ is odd). The sine terms for $k = 0$ and $k = 5$ vanish over $n = 0, \ldots, 9$ and are omitted.

The model's final prediction (i.e., $\mathbf{E}\mathbf{h}^L \in \mathbb{R}^{10}$) can be reconstructed via these six terms (see Appendix F), indicating that the final hidden state is encoded using Fourier bases.

Revisiting Figure 6, the first principal component (PC1) aligns with the parity vector $\boldsymbol{p}(n)$, separating even from odd digits. Second and third principal components span the $k = 2$ Fourier pair $(\cos, \sin(\frac{2\pi n}{5}))$, so the digits lie on two regular pentagons: one each for even and odd digits. The digits within each pentagon advance by $n + 4 \pmod{10}$ (e.g., $n = 0 \rightarrow 4 \rightarrow 8 \ldots$, same for odd digits), allowing a walk around the pentagon while staying within the even/odd set. Interestingly, taking $\pmod 5$ on such a sequence yields decreasing steps of 1 ($n \pmod 5 = 0 \rightarrow 4 \rightarrow 3 \ldots$). Lastly, the two pentagons are parallel and stacked along PC1, with corresponding vertices differing by $\pm 5$ (same phase, opposite parity). Together, these yield the pentagonal-prism geometry in Figure 6.

## 5 PITFALLS OF LEARNING: LACK OF LONG-RANGE DEPENDENCY

Given our insights, here we study the learning dynamics of the standard training model when learning multi-digit multiplication.

In particular, in Figure 7 (a), we inspect the gradient norms (top row) and losses (bottom row) *per token* $c_k$ over the course of training. There are a few observations to make.

First, note from the loss curves that the first two digits, $c_0, c_1$, followed by the last digit, $c_7$, are learned first, as indicated by their immediate drop in loss to near zero. This aligns with the gradient norms observed for these tokens: within the first few steps, these tokens receive gradients, but their norms quickly drop to near zero once the loss for these tokens reach zero. Also note that the order in which tokens are learned according to gradient norms and losses is consistent.

The model then eventually learns to predict $c_2$. However, middle digits, $c_3$ to $c_6$ are never learned. Despite only the middle digits receiving gradients (as they are the only sources of loss remaining), their losses plateau, suggesting that the model is stuck in a local optimum that lacks the long-range dependencies to properly learn the middle digits.

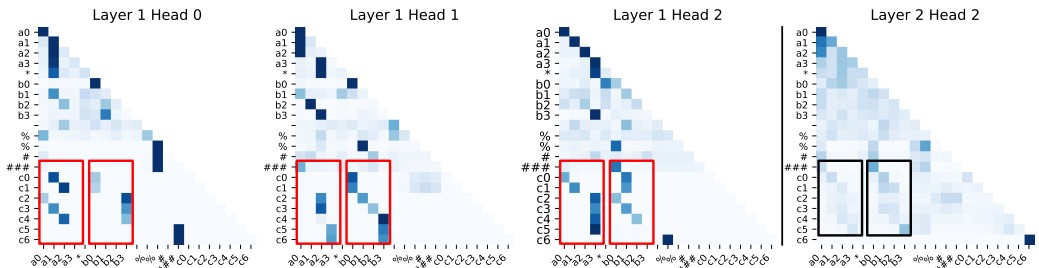

Figure 8: **Attention pattern of model trained with auxiliary loss.** This model similarly produces a "binary attention tree" (red boxes), but interestingly, we also see an attention head that attends to all necessary pairwise digits simultaneously (black box), producing a pattern akin to Figure 2.

Note that scaling to a larger model does not address this issue, as the same pattern can be found in a 12 layer 8 head model: see Appendix G.

# 6 LEARNING MULTIPLICATION WITHOUT ICoT OR 1-TO-N TRAINING

With our understandings thus far, we demonstrate an example of a simple fix to teach Transformers multiplications without needing ICoT or 1-to-N training.

In particular, we leverage the observation from Section 3.1 in that ($i$) multi-digit multiplication requires long-range dependencies between digit $c_k$ and pairwise products $\{a_i b_j | i + j \leq k\}$, and ($ii$) such dependency can be summarized by an intermediate value $\hat{c}_k$ to produce $c_k$.

Thus in order to guide the Transformer to learn long-range dependencies, we simply add an auxiliary loss term to predict $\hat{c}_k$ at each timestep $t_{c_k}$. We attach an additional linear regression head $\mathbf{w}_h \in \mathbb{R}^d$ to the output of $H(= 2)$ attention heads in the second layer. These regression heads are trained to predict the correct accumulated sum $\hat{c}_k$ at each timestep $t_{c_k}, c_k \in [0, \ldots 7]$ with a MSE loss:

$$z_i^h = \mathbf{w}_h^\top \text{ATT}_h^2(\cdot) \tag{7}$$

$$\mathcal{L}_{aux} = \frac{1}{H} \sum_{h \in H} \frac{1}{8} \sum_{i=0}^{7} (z_i^h - \hat{c}_i)^2 \tag{8}$$

$$\mathcal{L} = \mathcal{L}_{LM} + \lambda \mathcal{L}_{aux} \tag{9}$$

where $\mathcal{L}_{LM}$ is the standard language modeling loss.

This introduces an inductive bias for the task, and allows our 2-layer model to correctly learn 4x4 multiplication with 99% accuracy. Again, note that a larger 12-layer model still fails at multiplication under standard fine-tuning.

Revisiting Figure 7 (b) demonstrates a very different learning dynamic. We observe the model learn early and last digits ($c_0, c_1, c_7$) and work inwards ($c_2, c_3, c_4, c_6$, and finally $c_5$).

**Limitation.** Obviously the suggested inductive bias pertains specifically to our task. However, our experiments demonstrate the pitfall of Transformers that require long-range dependencies, and that it is possible to overcome such a pitfall with the correct inductive biases. We speculate that there are other generalizing inductive biases that can improve performance on tasks with long-range dependencies (Tay et al., 2020), and leave this for future work.

## 6.1 DOES THE MODEL WITH AUXILIARY LOSS LEARN THE SAME MECHANISMS?

One might ask whether ICoT, 1-to-N, and our auxiliary loss model learn the same mechanisms.

Inspecting the attention patterns suggests that a similar (but not necessarily exact) mechanism is learned: see Figure 8. Namely, the model similarly forms an "attention tree" to sparsely attend to the correct pairs of digits for each $c_i$ in the first layer (red boxes). Interestingly, in the auxiliary-loss model we also observe an attention head (Layer 2 Head 2) that simultaneously attends to all the

necessary digits, $\{a_{i \leq k}, b_{i \leq k}\}$, at each timestep $t_{c_k}$, forming a parallelogram-like attention pattern (black box) akin to the shape seen in Figure 2.

## 7 RELATED WORK

**Studying Transformers with Arithmetic Tasks.** A growing line of work study Transformers under controlled settings to better characterize their behavior (Allen-Zhu & Li, 2023a;b; Li et al., 2023; Nanda et al., 2023b; Park et al., 2024b;a). Often, arithmetics is a natural and popular domain (Lee et al., 2023; Ye et al., 2024; Nikankin et al., 2024), which has led to numerous insights. For instance, Nanda et al. (2023a) study how Transformers perform modular addition to explain grokking. Kantamneni & Tegmark (2025) find that large language models use trigonometry to do addition, encoding digits using Fourier bases, while Nikankin et al. (2024) suggest that they also rely on heuristics. Cai et al. (2025) study length generalization in Transformers using arithmetic tasks. Similarly, we study the limitations of Transformers by studying why it fails to learn multi-digit multiplication.

**Process Supervision.** Recent work trains models with *process supervision*, in which feedback is given not just on final correctness but on each intermediate reasoning step. For example, Uesato et al. (2022) demonstrate that process-supervision can yield less reasoning errors on GSM8K compared to outcome-only supervision. Similarly, Lightman et al. (2023) show that step-level human feedback on MATH leads to stronger reward models. More recently, Zhong et al. (2023)'s Math-Shepherd automates step-wise rewards via continuation-based verification, improving performance on both GSM8K and MATH. ICoT similarly plays the role of process supervision in latent space, by slowly removing chain-of-thought tokens during training such that the model internalizes the reasoning procedure. We thus use ICoT's success on multiplication to study why Transformers fail.

## 8 CONCLUSION

In this work, we study why Transformers fail on a seemingly simple task of multi-digit multiplication. We answer this question by reverse-engineering models trained with either implicit chain-of-thought or a 1-to-N training regime, and uncover that successful models learn the correct long-range dependencies needed for multi-digit multiplication. Our findings point to a pitfall of the standard recipe for training language models: using gradient descent with an auto-regressive loss on Transformers does not encourage the model to learn the right long-range dependencies. While we provide a simple example of how the right inductive bias can address such a limitation, we anticipate future work to provide a generic solution to improve on tasks with long-range dependencies.

### REPRODUCIBILITY STATEMENT

Our code to reproduce all of our experiments can be found in `https://anonymous.4open.science/r/icot-F822/`. Appendix A provides details of our training setup, including data formats, sample size, and hyperparameters.

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

# A    TRAINING DETAILS

Here we provide additional details regarding the training procedures of each of our models.

## A.1    ICoT TRAINING

Our ICoT training setup follows the practice outlined in the original ICoT paper (Deng et al., 2024).

ICoT works by initially presenting explicit chain-of-thought tokens during training, but gradually removing them across numerous "stages" (e.g., epochs). Concretely, the training examples at each epoch may have the following form:

$$\text{(Epoch 1)} \quad a_0a_1a_2a_3 * b_0b_1b_2b_3 \%\%\% \; q_0 \dots q_i \dots q_j \dots q_k \dots q_\tau \; \#\#\#\# \; c_0 \dots c_7$$
$$\text{(Epoch 2)} \quad a_0a_1a_2a_3 * b_0b_1b_2b_3 \%\%\% \; q_i \dots q_j \dots q_k \dots q_\tau \; \#\#\#\# \; c_0 \dots c_7$$
$$\text{(Epoch 3)} \quad a_0a_1a_2a_3 * b_0b_1b_2b_3 \%\%\% \; q_j \dots q_k \dots q_\tau \; \#\#\#\# \; c_0 \dots c_7$$
$$\dots$$
$$\text{(Epoch N)} \quad a_0a_1a_2a_3 * b_0b_1b_2b_3 \%\%\% \; \#\#\#\# \; c_0 \dots c_7$$

where $q_i$ are CoT tokens and $\%, \#$ are special delimiters. These delimiters have no special meaning beyond matching the setup of Deng et al. (2024). Note that after each epoch, the model sees a shorter chain by truncating some tokens, and that by the end, only the operands and final answer remain.

The actual format of our ICoT data is as follows. Using an example input of $8331 \times 5015$, digits are presented in least-significant digits first, resulting in the following format:

$$1338 * 5105 || 55614 + 013380(569421) + 0000000(5694210) + 0005561\%\%\#\#\#\#56997714$$

Unlike Deng et al. (2024), instead of using a pre-trained 12-layer GPT model, we train a smaller 2-layer, 4-head GPT-based model from scratch, not only to remove any confounding factors from pre-trained knowledge, but also because the 2-layer 4-head architecture is the simplest form in which ICoT succeeds but standard fine-tuning fails. The training data consists of 80,800 samples, while the validation and test sets each contain 1,000 held out samples. We train with a learning rate of `5e-5`, and remove 8 chain-of-thought tokens at every "stage" (which in our case is an epoch). Both training and validation loss converge after 13 epochs, and achieves 100% accuracy on the test set.

## A.2    1-TO-N TRAINING

An alternative way to train a Transformer to learn $4 \times 4$ digit multiplication is to present smaller digit multiplications in the training data. Namely, as opposed to providing only $4 \times 4$ digit multiplication, we present samples of (1-to-N) $\times$ (1-to-N) digit multiplication as well (i.e., $1 \times 4$, $2 \times 4$, $3 \times 4$ digit multiplication, but also $1 \times 2$, $1 \times 3$, $2 \times 2$ etc.), reflecting the strategy introduced in Shen et al. (2023). We call this training regime "1-to-N training".

Similar to ICoT, we train a 2-layer 4-head decoder-only model from scratch. We train on 10 million samples for 10 epochs, with a learning rate of $1e-4$, and use the same validation set as the ICoT model. Our 1-to-N model reaches 99.9% accuracy.

## A.3    STANDARD FINE-TUNING

Similar to ICoT, for our standard fine-tuning model, we train a 2-layer, 4-head GPT-based model from scratch, on the same data as ICoT. We use a learning rate of `5e-5`, and the input format is $a_0a_1a_2a_3 * b_0b_1b_2b_3 \%\%\#\#\#\#c_0 \dots c_7$. All other hyperparameters match those in our ICoT setup. The model's loss and accuracy plateaus after 13 epochs, it achieves only about 1% train and validation accuracy, while digit-level accuracy converges at approximately 81%, and remains the same even after 60 epochs.

Note that scaling the model larger to a 12-layer, 8-head model achieves the same low accuracy at 1% and digit-level accuracy of 80%.

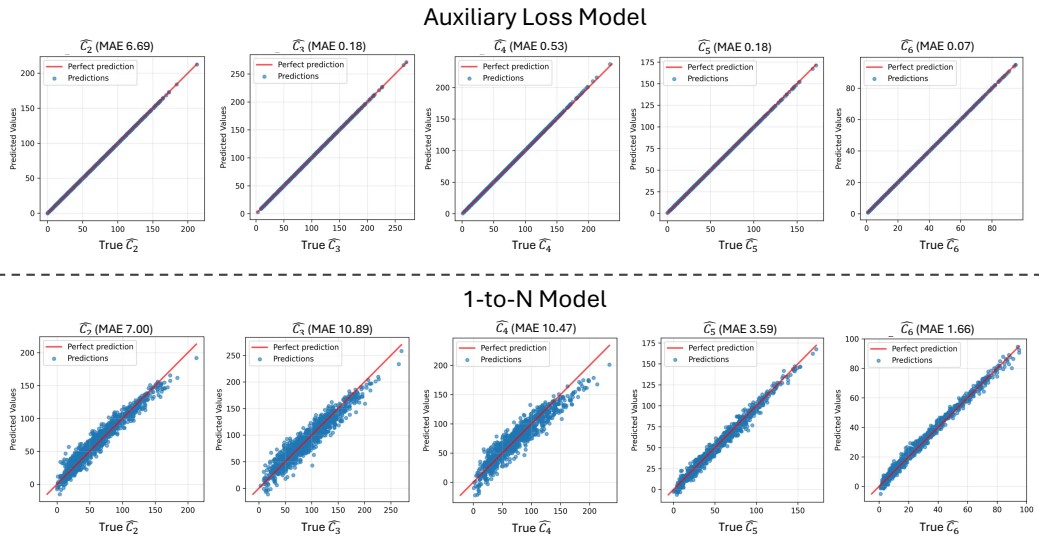

Figure 9: **Linear regression probing results for $\hat{c}$, for 1-to-N and Auxiliary Loss models.** We probe from the middle of the last Transformer block, after attention heads but before MLPs.

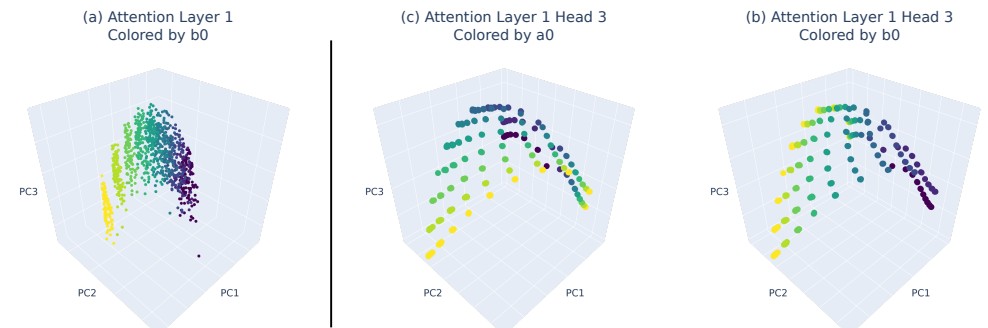

Figure 10: **3D PCA of attention head outputs from the auxiliary loss model.**

## B ADDITIONAL $\hat{c}$ PROBING RESULTS

Here we provide probing results for $\hat{c}$ for the 1-to-N model and the auxiliary loss model: see Figure 9. The auxiliary loss model has near perfect probing accuracy, as expected since it was directly trained to do so, while the 1-to-N model has a much lower accuracy, even compared to the ICoT model.

## C MORE EXAMPLES OF ATTENTION HEAD OUTPUTS

In Section 4.1 we show examples of attention head outputs in the ICoT model. Here we demonstrate more examples from the 1-to-N and auxiliary loss models: see Figure 10, 11.

## D NESTED REPRESENTATIONS

Here we describe how nested representations as seen in Figure 5 can emerge under PCA.

As a reminder, we use standard letters $(x, y, \alpha)$ for scalars, bold letters for vectors $(\mathbf{a}_x, \mathbf{b}_y \in \mathbb{R}^d)$, and calligraphics for sets $(\mathcal{X}, \mathcal{Y} = \{0, ..., 9\})$. $a_{0,...,3}, b_{0,...,3}$ refer to positions of digits in the two operands. Let $x \in \mathcal{X} := \{0, ..., 9\}, y \in \mathcal{Y} := \{0, ..., 9\}$ be the actual digit values at positions $a_i, b_j$.

Consider the case in which an attention head only attends to two positions, $a_i, b_j$ (a digit in the first operand, another digit in the second operand), as we often observe for the ICoT model. Let $W_O \in$

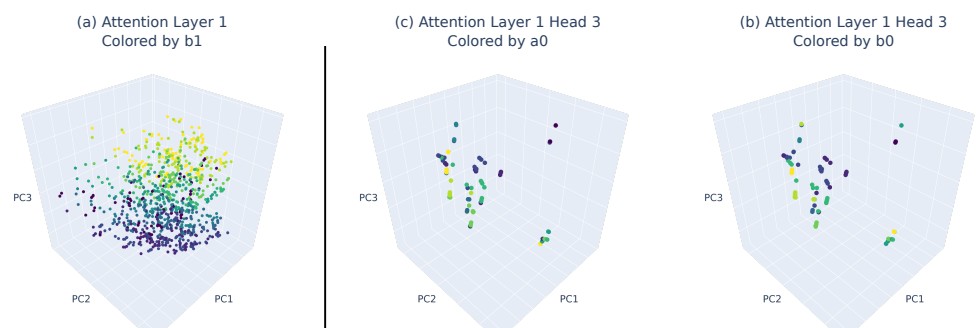

Figure 11: **3D PCA of attention head outputs from the "1-to-N" model.**

$\mathbb{R}^{d \times d_{head}}, W_V \in \mathbb{R}^{d_{head} \times d}$ be the head's output and value weights. Let $E[x], E[y] \in \mathbb{R}^d$ be the token embeddings of digits $x, y$. Finally, let $\mathbf{a}_x = W_O W_V E[x], \mathbf{b}_y = W_O W_V E[y], \mathbf{a}_x, \mathbf{b}_y \in \mathbb{R}^d$.

When the attention head only attends to two positions, with possible values $x \in \mathcal{X}, y \in \mathcal{Y}$, the output of the attention head can be expressed as

$$\mathbf{z}_{xy} := \alpha \mathbf{a}_x + (1 - \alpha) \mathbf{b}_y \tag{10}$$

(ignoring position embeddings) where $\alpha$ indicates how much attention is dedicated to position $a_i$. For simplicity, we assume a fixed constant $\alpha$ value.

Henceforth, we abuse notations and let $\mathbf{a}_x, \mathbf{b}_y, \mathbf{z}_{xy}$ each refer to their mean centered counterparts:

$$\mathbf{a}_x := \mathbf{a}_x - \mathop{\mathbb{E}}_{x \in \mathcal{X}}[\mathbf{a}_x], \qquad \mathbf{b}_y := \mathbf{b}_y - \mathop{\mathbb{E}}_{y \in \mathcal{Y}}[\mathbf{b}_y], \qquad \mathbf{z}_{xy} := \mathbf{z}_{xy} - \mathop{\mathbb{E}}_{x \in \mathcal{X}, y \in \mathcal{Y}}[\mathbf{z}_{xy}] \tag{11}$$

To see why the nested structure in Figure 5 shows up with PCA, we can inspect the covariance of the attention head's output. Define covariances of $\mathbf{a}_x, \mathbf{b}_y$ as

$$\Sigma_{\mathbf{a}} := \mathop{\mathbb{E}}_{x \in \mathcal{X}}\left[\mathbf{a}_x \mathbf{a}_x^\top\right], \quad \Sigma_{\mathbf{b}} := \mathop{\mathbb{E}}_{y \in \mathcal{Y}}\left[\mathbf{b}_y \mathbf{b}_y^\top\right], \tag{12}$$

First, we show that the covariance of the model's output $\Sigma_{\mathbf{z}} = \alpha^2 \Sigma_{\mathbf{a}} + (1 - \alpha)^2 \Sigma_{\mathbf{b}}$:

$$\Sigma_{\mathbf{z}} = \mathop{\mathbb{E}}_{x \in \mathcal{X}, y \in \mathcal{Y}}\left[\mathbf{z}_{xy} \mathbf{z}_{xy}^\top\right] = \mathop{\mathbb{E}}_{x \in \mathcal{X}, y \in \mathcal{Y}}\left[\left(\alpha \mathbf{a}_x + (1 - \alpha) \mathbf{b}_y\right)\left(\alpha \mathbf{a}_x + (1 - \alpha) \mathbf{b}_y\right)^\top\right] \tag{13}$$

$$= \alpha^2 \mathop{\mathbb{E}}_{x \in \mathcal{X}}[\mathbf{a}_x \mathbf{a}_x^\top] + (1 - \alpha)^2 \mathop{\mathbb{E}}_{y \in \mathcal{Y}}[\mathbf{b}_y \mathbf{b}_y^\top] + \alpha(1 - \alpha) \mathop{\mathbb{E}}_{x \in \mathcal{X}, y \in \mathcal{Y}}[\mathbf{a}_x \mathbf{b}_y^\top] + \alpha(1 - \alpha) \mathop{\mathbb{E}}_{x \in \mathcal{X}, y \in \mathcal{Y}}[\mathbf{b}_y \mathbf{a}_x^\top]$$

$$\tag{14}$$

Note that $\mathcal{X}, \mathcal{Y}$ are independent, and $\mathbb{E}[\mathbf{a}_x], \mathbb{E}[\mathbf{b}_y] = 0$, thus

$$\mathbb{E}[\mathbf{a}_x \mathbf{b}_y^\top] = \mathbb{E}[\mathbf{a}_x]\mathbb{E}[\mathbf{b}_y]^\top = \mathbf{0}, \quad \mathbb{E}[\mathbf{b}_y \mathbf{a}_x^\top] = \mathbf{0}. \tag{15}$$

Thus the cross terms vanish and we are left with

$$\Sigma_{\mathbf{z}} = \alpha^2 \mathop{\mathbb{E}}_{x \in \mathcal{X}}[\mathbf{a}_x \mathbf{a}_x^\top] + (1 - \alpha)^2 \mathop{\mathbb{E}}_{y \in \mathcal{Y}}[\mathbf{b}_y \mathbf{b}_y^\top] \tag{16}$$

$$= \alpha^2 \Sigma_{\mathbf{a}} + (1 - \alpha)^2 \Sigma_{\mathbf{b}}. \tag{17}$$

Importantly, note that $\Sigma_{\mathbf{a}}, \Sigma_{\mathbf{b}}$ are each derived from $\mathbf{a}, \mathbf{b}$, which are both obtained by applying the same linear map $W_O, W_V$ to digit embeddings $E[\cdot]$. Furthermore, the two digits at the two positions being attended to are both sampled from a uniform distribution over $\{0, \ldots, 9\}$, and thus the marginal distributions of $\mathbf{a}_x$ and $\mathbf{b}_y$ are essentially the same. Under such a setting we have $\Sigma_{\mathbf{a}} \approx \Sigma_{\mathbf{b}}$ and thus they have (approximately) the same eigenvectors.

This also extends to $\Sigma_{\mathbf{z}}$:

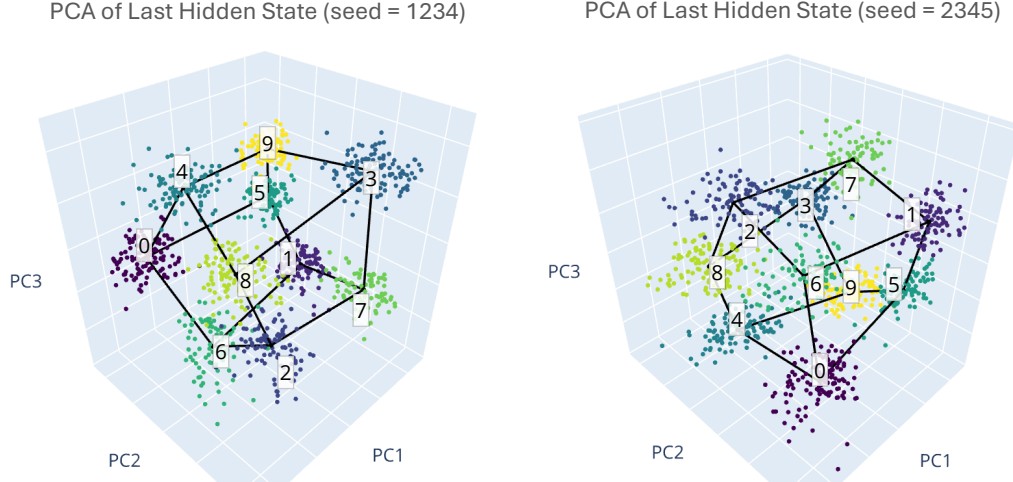

Figure 12: **3D PCAs of the final layers of the ICoT model, using different random seeds.** This structure consistently shows up across training runs.

$$\Sigma_{\mathbf{a}} = U\Lambda_{\mathbf{a}}U^{\top}, \quad \Sigma_{\mathbf{b}} = U\Lambda_{\mathbf{b}}U^{\top}, \tag{18}$$

$$\Sigma_{\mathbf{z}} = \alpha^2\Sigma_{\mathbf{a}} + (1-\alpha)^2\Sigma_{\mathbf{b}} = U\big(\alpha^2\Lambda_{\mathbf{a}} + (1-\alpha)^2\Lambda_{\mathbf{b}}\big)U^{\top} \tag{19}$$

and $\Sigma_{\mathbf{z}}$ also shares eigenvectors with $\Sigma_{\mathbf{a}}, \Sigma_{\mathbf{b}}$. Thus the principal components picked by PCA for the attention head outputs (i.e., eigenvectors of $\Sigma_{\mathbf{z}}$, i.e., "global geometry") are the same as $\Sigma_{\mathbf{a}}$, and $\Sigma_{\mathbf{b}}$.

Now we focus on the "local" geometry of the attention head's output. To do so, assume we fix the value of $x$ as $x_0$. The set of outputs of the attention head is $\{\mathbf{z}_{xy}|x = x_0\} = \{\alpha\mathbf{a}_{x_0} + (1-\alpha)\mathbf{b}_y\}_{y\in\mathcal{Y}}$. The local covariance of $\{\mathbf{z}_{xy}|x = x_0\}$ is then $\Sigma_{x=x_0} := (1-\alpha)^2\Sigma_{\mathbf{b}}$, as $\Sigma_{\mathbf{a}}$ goes to zero when $\mathbf{a}_{x_0}$ is a constant once $x$ is fixed. Thus, $\Sigma_{x=x_0}$ shares the same eigenvectors of $\Sigma_{\mathbf{b}}$, with its eigenvalues simply being scaled by $(1-\alpha)^2$.

To summarize, when applying a 3D PCA projection to the attention head's output, the "global" principal components are equal to the eigenvectors of $\Sigma_{\mathbf{z}}$. Under our setup, these coincide with those of $\Sigma_{\mathbf{a}}$ and $\Sigma_{\mathbf{b}}$. When we fix the value of $x$ or $y$, the local covariance becomes a scalar multiple of $\Sigma_{\mathbf{a}}$ or $\Sigma_{\mathbf{b}}$, so each cluster of points (set of points in the box of Figure 5 (b)) is organized along the same principal directions. The shared eigenvectors between global and local covariances is what produces the nested PCA visualizations.

## E    MORE EXAMPLES OF 3D PCA OF HIDDEN STATES

In Section 4.2 we show how the ICoT model's final layer is encoded in a Fourier basis. With ICoT, this geometry consistently shows up across training runs with random seeds (see Figure 12.

On the other hand, the 1-to-N model and auxiliary loss models do not, despite the fact that both models have learned multi-digit multiplication: see Figure 13. Unlike the ICoT model, we do not observe a clear Fourier pattern.

## F    FOURIER STRUCTURE IN MODEL'S WEIGHTS, ACTIVATIONS

Here we provide a deeper dive into the Fourier structure found in the ICoT model's weights and hidden states. Namely, we analyze the model's embedding weights, final MLP's weights, and last hidden layer:

1. Embeddings $\mathcal{E} \in \mathbb{R}^{10\times d}$
2. Final layer MLP output weights $W_{out} \in \mathbb{R}^{d_{mlp}\times d}$, given $\text{MLP}(\mathbf{x}) = \sigma(W_{in}\mathbf{x})W_{out}$

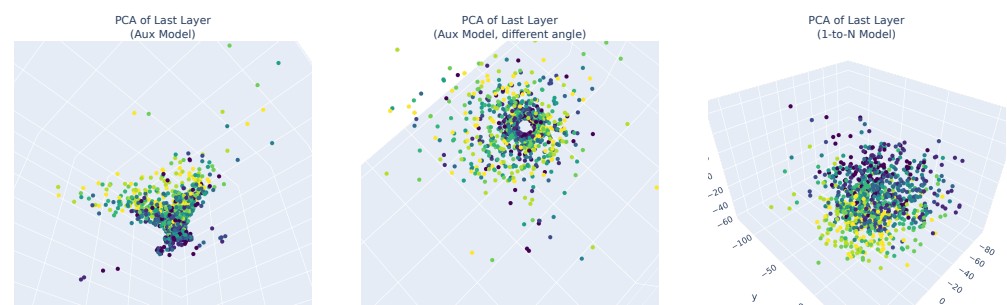

**Figure 13: 3D PCAs of the final layers of the auxiliary loss model and 1-to-N model.** Though there are some noticeable patterns, we do not observe a clear Fourier structure as in the ICoT model.

    3. Final hidden layer $\mathbf{h}_t^L \in \mathbb{R}^{N \times d}$

where $N(=1,000)$ is the size of our validation set. For the latter two, we first project them onto the model's embedding space:

$$\widehat{W_{out}} = (\mathcal{E} W_{out})^\top \in \mathbb{R}^{d_{mlp} \times 10}$$

$$\widehat{\mathbf{h}^L} = (\mathcal{E} \mathbf{h}^L)^\top \in \mathbb{R}^{N \times 10}$$

Each item $X \in \{\mathcal{E}, \widehat{W_{out}}, \widehat{\mathbf{h}^L}\}$ is a collection of row vectors $\mathbf{x} \in \mathbb{R}^{10}$ whose ten entries correspond to digits $n = 0, \ldots, 9$.

We find that vectors $\mathbf{x}$ are encoded in a low-dimensional trigonometric subspace.

Namely, consider the Fourier expansion

$$\sum C_n e^{-2\pi i \frac{kn}{10}}, \quad n = 0, \ldots, 9.$$

where $C_n (\neq c_k)$ is some constant per $n$. Following Kantamneni & Tegmark (2025), we take frequencies $k \in \{0, 1, 2, 5\}$, yielding the real Fourier basis

$$\Phi(n) = \left[ \begin{array}{cccccc} \underset{(k=0)}{\mathbf{1}(n)} & \underset{(k=1)}{\cos\left(2\pi \frac{n}{10}\right)} & \underset{(k=1)}{\sin\left(2\pi \frac{n}{10}\right)} & \underset{(k=2)}{\cos\left(2\pi \frac{n}{5}\right)} & \underset{(k=2)}{\sin\left(2\pi \frac{n}{5}\right)} & \underset{(k=5)}{\boldsymbol{p}(n)} \end{array} \right],$$

where $\mathbf{1}(n) \equiv 1$ (the DC component) and $\boldsymbol{p}(n) \equiv (-1)^n$ (the Nyquist/parity vector). The sine terms for $k = 0$ and $k = 5$ vanish over $n = 0, \ldots, 9$ and are omitted.

Let $F \in 10 \times 6$ be a Fourier matrix with rows indexed by $n \in \{0, \ldots, 9\}$ and columns as defined above.

For each row $\mathbf{x} \in \mathbb{R}^{10}$ we fit least squares coefficients

$$C = \arg\min_{C \in \mathbb{R}^6} \|x - FC\|_2^2$$

and quantify goodness-of-fit using coefficient of determination

$$R^2(x) = 1 - \frac{\|x - FC\|_2^2}{\|x - \bar{x}\|_2^2},$$

We report the median $R^2$ over the set of rows in each $X$ (i.e., over $d_{\text{mlp}}$ rows for $\widehat{W_{out}}$, $d$ rows for $\mathcal{E}$, and over batch examples for $\mathbf{h}^L$).

In Table 1 we observe strong fits: the per-row medians lie between $0.85$ and $0.99$, indicating that a six-dimensional trigonometric basis over digits captures the vast majority of variance:

We can extend the Fourier bases to include additional terms, for $k = 3, 4$, which forms a 8 dimensional basis (excluding sine terms for $k = 0, 5$), which leads to perfect $R^2$ fits.

Table 1: Median $R^2$ of Fourier fits over digits ($n = 0 \ldots 9$).

| Object | Fourier Basis | Rows aggregated | Median $R^2$ |
|---|---|---|---|
| $\mathcal{E}$ | $k = 0, 1, 2, 5$ | $d_{\text{model}}$ | 0.84 |
| MLP $W_{out}$ weights | $k = 0, 1, 2, 5$ | $d_{\text{mlp}}$ | 0.95 |
| $\mathbf{h}^L$ | $k = 0, 1, 2, 5$ | batch examples | 0.99 |
| $\mathcal{E}$ | $k = 0, 1, 2, 3, 4, 5$ | $d_{\text{model}}$ | 1 |
| MLP $W_{out}$ weights | $k = 0, 1, 2, 3, 4, 5$ | $d_{\text{mlp}}$ | 1 |
| $\mathbf{h}^L$ | $k = 0, 1, 2, 3, 4, 5$ | batch examples | 1 |

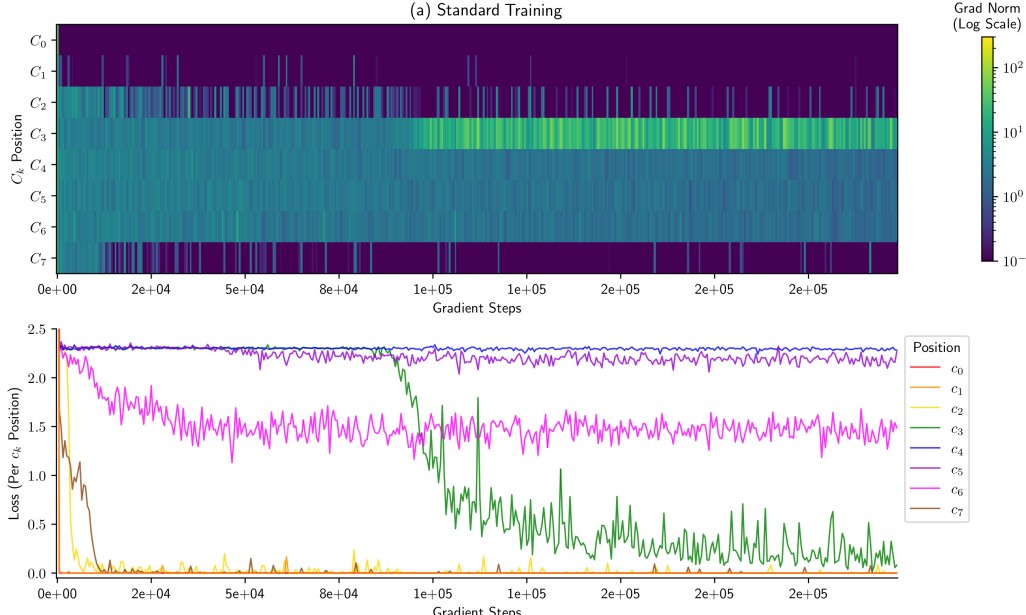

Figure 14: **Gradients and loss per token for a 12-layer model.**

## G PER TOKEN GRADIENTS AND LOSSES: 12-LAYER MODEL

Even with a larger 12 layer model, the model fails to learn the right long-range dependencies. Figure 14 displays the results – we see the similar patterns as the 2-layer model, in which middle digits never receive the right gradients and loss does not drop.

## H ATTENTION PATTERNS OF ALL MODELS

In Section 3.3, we illustrate how a binary tree is constructed for $c_2$ in the ICoT model. In Figure 15 and Figure 16, we present the attention patterns for all digits across the four models, from with attention trees can be derived for the ICoT model for each solution token $c_i$.

## I LLM USAGE

We used LLMs to proof read our draft and polish our notations.

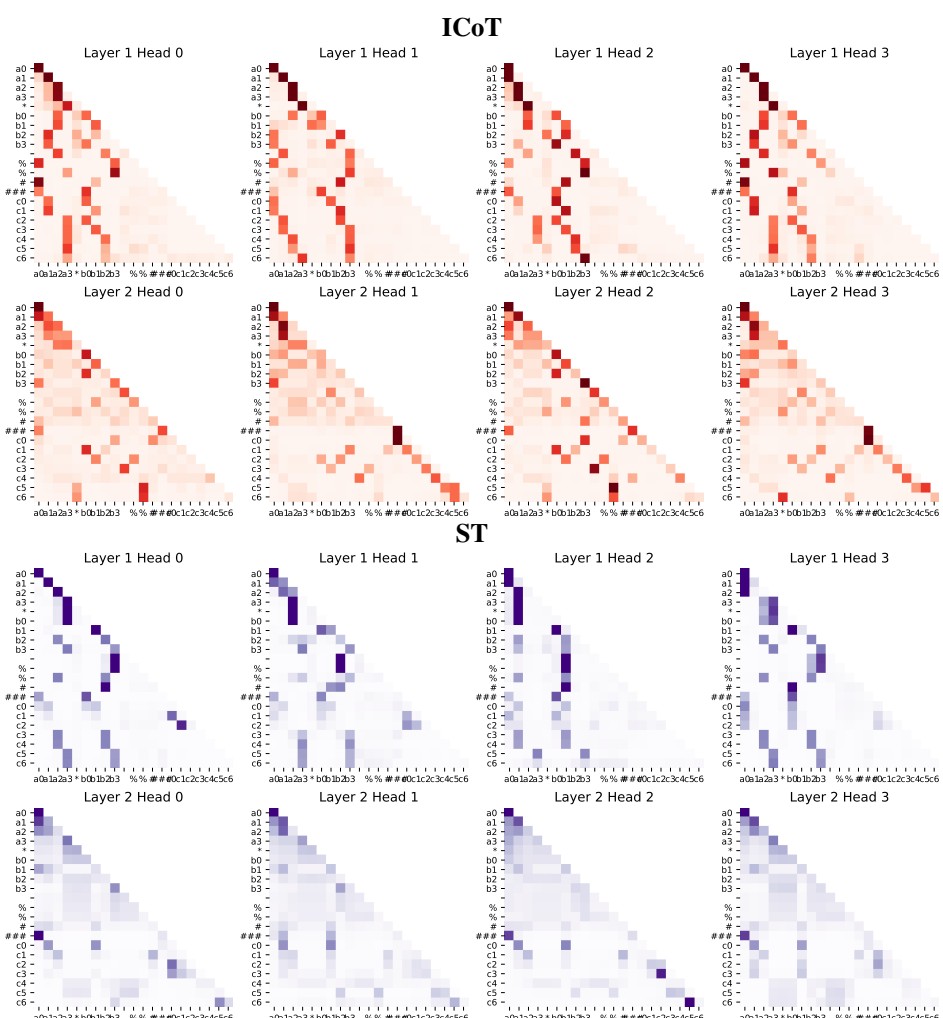

Figure 15: Attention patterns of ICoT and standard fine-tuned model

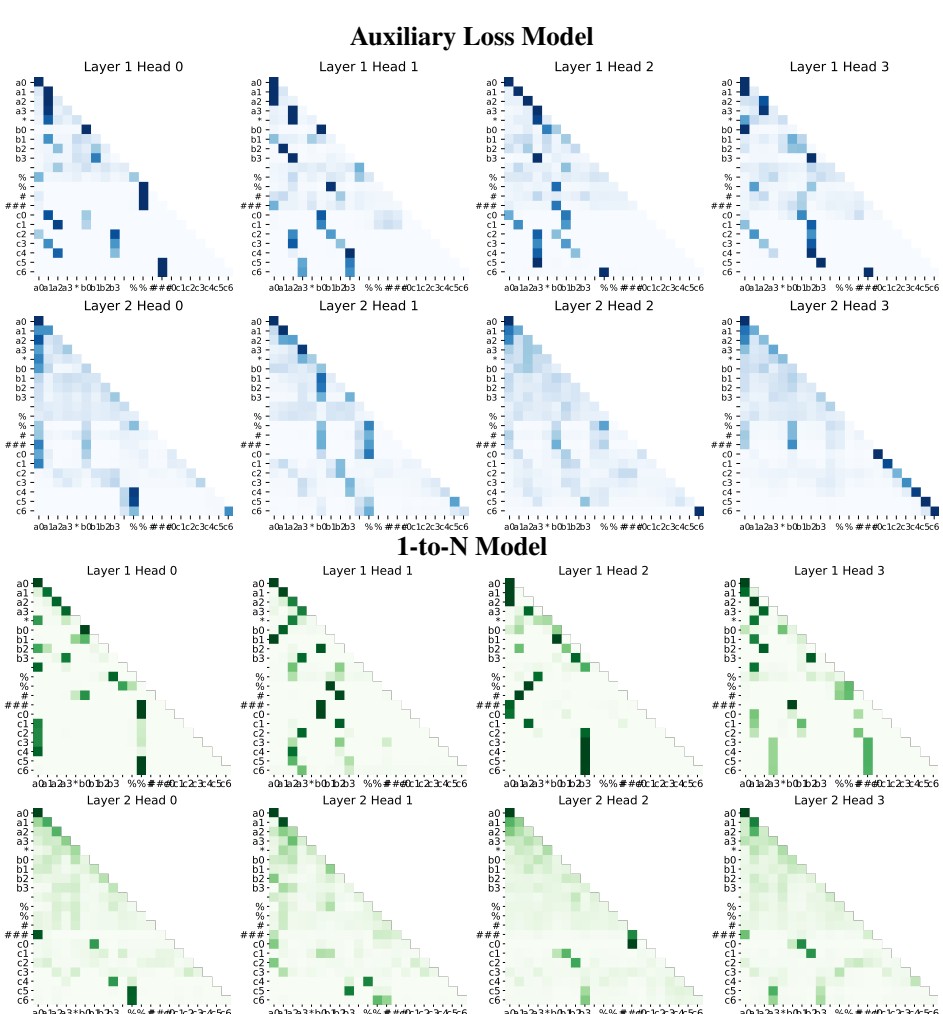

Figure 16: Attention patterns of the model trained with auxiliary loss and the 1-to-N model

