# OpenReview forum: "Why Can't Transformers Learn Multiplication? Reverse-Engineering Implicit Chain-of-Thought Reveals Challenges of Learning Long-Range Dependencies"
_ICLR.cc/2026/Conference — Submitted to ICLR 2026_

### Official Review · Reviewer_onsG · 2025-10-22

**Soundness:** 4
**Presentation:** 4
**Contribution:** 3
**Rating:** 10
**Confidence:** 4

**Summary:**

This paper investigates why Transformers fail at multi-digit multiplication. The authors reverse-engineer an implicit chain-of-thought (ICoT) model that successfully learns 4×4 multiplication and compare it to a standard fine-tuned model that fails. Through mechanistic analysis, they discover that the ICoT model encodes necessary long-range dependencies by constructing a binary-tree-like attention pattern that caches pairwise partial products across timesteps for later retrieval. The model also uses sophisticated geometric representations, encoding digits with Fourier bases that form a pentagonal prism structure.

In contrast, standard fine-tuning gets stuck in a local optimum where only the first, last, and some early digits are learned correctly - the model fails to learn the middle digits that require tracking complex dependencies. To validate their understanding, the authors introduce an auxiliary loss that supervises the model to predict intermediate "running sums," providing the inductive bias needed to achieve 99% accuracy without chain-of-thought supervision. This work reveals a fundamental challenge in how Transformers learn long-range dependencies under standard training and demonstrates that while task-specific fixes exist, more general solutions are needed.

**Strengths:**

This is an excellent paper.

The mechanistic investigation of the ICOT model builds up over several layers, from theoretical analysis of long-range dependencies, demonstrated through logit attributions, linear regression probing, use of some attention heads to handle long-range dependencies, the outputs of some attention heads forming a Minkowski sum, a 3D two-level PCA analysis, and finally a Fourier analysis.

The pentagram prism structure, revealed in the Fourier analysis, is a novel insight into how the models encodes information about our base 10 counting system. This “tool-like” structure appears to embed: A) 10 is divisible by 5 and 2 B) concept of odd and even numbers C) potentially a way to calculate n+5 and n-5 (between layers) and D) n+4 and n-4 (around rings).

The Figures are excellent - particularly 1 & 4 as they represent hard-to-diagram concepts. Figure 6 is beautifully presented - the pentagon prism gives a really great insight into how this model represents base 10 digits in model space.

The ICOT model insights lead directly to a modified loss function allowing SFT of a model to perform multiplication accurately. This strengthens the case for the ICOT insights being valid.

**Weaknesses:**

None to speak of. I think the Conclusion undersells what the paper achieves.

Minor points:
- Minkowski sums are first used without explanation. Consider adding “(adding sets of vectors geometrically)”.
- Section 3.3, the notation “ATT superscript 2 subscript 3” is not explained before use, and differs from the “Layer 2 Head 3” notation used in Figure 4. Later “ATT superscript 1 (i, j)” is used with little introduction. Consider standardizing in some way to make reading easier.
- Figure 5, consider swapping position of 2nd and 3rd image for easier left to right reading- making the zoomed-out and zoomed-in images sit side by side.

**Questions:**

Q1: The pentagon prism structure appears to embed many facts about our base 10 counting system. Do you expect this tool-like structure to have been independently learnt by other models trained on base 10 math?

Q2: The pentagon prism has a 5 ring and 2 layers. The primes of 10 are 5 and 2. If the model had been trained in base 15, what shaped structure would you expect?

---

> ### Author Response · Authors · 2025-11-21
>
> Thank you for your time, careful review, and excitement for our paper! We appreciate you recognizing **the thoroughness of our mechanistic investigation**, **our novel insights into how the model encodes the counting system**, and **the clarity of our presentation**.
>
> **Minor point 1: Clarification on Minkowski Sums**: Per comments from reviewer cmTc, we will make our notations regarding Minkowski sums more precise, including what it means to add two sets of vectors together.
>
> **Minor point 2: Clarification Notions**: Thank you for catching this - we will make our notations of ATT consistent throughout the text.
>
> **Minor point 3: Figure Clarity**: I think this is a sensible choice - thank you! We have updated Figure 5.
>
> **Q1: How the Geometric Structure Generalizes to Other Bases**: These are fun questions! Yes, we speculate that similar structures can be found in other models trained on base 10 math, as we have already seen more evidence that models often encode numbers using Fourier bases [1, 2].
>
> **Q2**: Certainly the prime factors of 10 (2, 5) likely show up in the first 3 principal components, because it’s a very simple way for the model to compress 10-digit numbers (by combining mod 2 and mod 5). By extension, for base 15, we think it is completely reasonable to similarly expect a decomposition of numbers into a triangle (i.e., 3-cycle aka mod 3) and pentagon (i.e., “5-cycle” aka mod 5). One possible way to visualize this in 3 dimensions may be to think of a pentagon repeated across 3 layers or a triangle repeated across 5 layers. To reviewer CsfC’s point, another interesting base would be base 11, which is already a prime number. With no factorization available, our guess is that the model would represent the 11 digits in a single cycle.
>
> [1] Kantamneni et al., “Language Models Use Trigonometry to Do Addition” 2025
> [2] Nanda et al., “Progress measures for grokking via mechanistic interpretability”, 2023

---

### Official Review · Reviewer_cmTc · 2025-10-31

**Soundness:** 2
**Presentation:** 1
**Contribution:** 2
**Rating:** 2
**Confidence:** 4

**Summary:**

This paper investigates how Transformers acquire long-range dependencies for multiplication via an implicit chain of thought (ICoT). Linear probing analyses indicate that ICoT encourages Transformer attention to cache pairwise products, providing evidence of long-range dependency capture. A 3D mapping of attention head outputs reveals nested representations of number tokens. The authors also propose a loss function that enables Transformers to learn multiplication efficiently without ICoT.

**Strengths:**

- The paper thoroughly examines how a Transformer with ICoT processes input digits to perform multiplication, supported by extensive experiments.
- Linear probing shows that the attention mechanism indeed stores intermediate computations.
- The geometric analysis reveals an interesting pentagram-like structure.

**Weaknesses:**

The paper has unclear points in several critical areas (premise, definitions, and mathematical exposition), which made it difficult to fully appreciate the contributions.

---
**Premise and baseline model.**

First, while the authors claim that SFT models cannot learn multiplication, prior work reports that simple GPT models readily learn multiplication when using zero-padding (to fix the number of digits) and digit reversal. Indeed, [1] shows that GPT-2 small successfully learns multiplication up to 15×15 digits. The present paper appears to use the same technique, which was not used in [Yang et al., 2023]—the work cited as evidence that multiplication is hard to learn. Consequently, the premise of this paper seems to rest on an unfair contrast: the claimed hardness relies on [Yang et al., 2023], whereas the technique employed follows [1], which already addresses the hardness.

[1] Shen et al., "Positional Description Matters for Transformers Arithmetic," 2023.

Second, it is unclear what “SFT models” refers to. No precise definition is given. Line 71 only states “a model trained with standard fine-tuning,” but if it is fine-tuned, on what pretraining data? What is the architecture—encoder–decoder or decoder-only? In either case, why does the baseline fail to learn multiplication unlike in [1]? Reversing target digits already encourages a certain chain of thought. The baseline should be able to reach near-perfect accuracy on the target task (i.e., 4×4-digit multiplication).

---

**Presentation**

The presentation obscures several core results.

The main text states that Figure 3 shows mean absolute error, but the scatter plots do not appear to reflect that. What are the diagonal points? In the upper row, why are blue points clustered at the bottom right?

Section 4 is also difficult to follow due to many undefined or cluttered symbols and operators.
- [Eq. 4] $A, B, \oplus$ are undefined. If $A$ is a matrix, the Minkowski sum between a matrix and a scalar $\epsilon$ is not defined. Assuming broadcasting, the right-hand side is a matrix while the left-hand side is a set, so the statement “matrix includes a set” is not meaningful.
- [Eq. 5] $\mathrm{Cov}(A_i)$ is undefined. Is this the covariance over entries in $A_i$, or across the index $i=1,2,\dots$? It is also unclear why $\Sigma_A$ and $\Sigma_B$ should share eigenvectors. The “local” covariance $\Sigma_{\mathrm{local}\mid a_i}$ appears to be conditioned on $a_i$, but its definition is not.
- [Fourier expansion] The index $k$ and the operator $*$ are undefined. $\mathbb{1}(n)$ and $\mathbb{p}(n)$ are not defined; the notation suggests vectors, but other terms are not vectors, which is inconsistent. What does $\mathbb{1}(n) \equiv 1$ mean?

---
**Insights**

While the paper aims to explain why multiplication is hard for SFT but feasible with ICoT, the results do not fully support this claim. Linear probing shows that how the successful model (i.e., the ICoT model) processes the digits while the SFT model does not. This finding is interesting but describes what happens in successful versus unsuccessful models rather than explaining what causes success or failure. Section 5 states:

> Despite only the middle digits receiving gradients (as they are the only sources of loss remaining), their losses plateau, suggesting that the model is stuck in a local optimum that lacks the long-range dependencies to properly learn the middle digits.

However, attributing failure to being stuck in a local optimum—without theoretical justification—does not provide a satisfactory explanation.

---

Overall, the work appears to rest on an incorrect (or unfair) premise that SFT models cannot learn even 4×4-digit multiplication, thereby overestimating the task’s hardness and overstating the benefits of ICoT. In addition, the presentation requires substantial improvements. Finally, the claim to explain “why” should be made with greater care to avoid overstatement.

**Questions:**

Please address the weaknesses raised above.

---

> ### Author Response · Authors · 2025-11-21
>
> Thank you for your careful review and for finding the geometry analysis interesting.
>
> ### **Concern about Unclear and Potentially Unfair Baseline Definition**
>
> Thank you for pointing us to [1]. We have missed this work, and indeed, their work suggests that transformers can learn multi-digit multiplication. We can see how this can cause confusion and that our claims seem to contradict theirs. We took a closer look and found the following:
>
> Although we do employ one of their techniques (reversing digit order), as it turns out, this technique alone is not sufficient to get the Transformer to learn multiplication. There’s an additional subtle difference in their setup. In particular, the training data that we use always has the form of 4x4-digit multiplication. In [1], their training data consists of “1-to-N x 1-to-N” digit multiplication (e.g., 1x4 digit multiplication, 2x4, 3x4, 4x4, but also 2x2, 2x3, etc.).
>
> As it turns out, with this “1-to-N training”, we are able to reach 99.9% accuracy in a 2 layer 4 head model.
>
> With that being said, we will make changes to our narrative, as to your point, our current claims can be misleading. In particular, one might characterize both “1-to-N training” and ICoT as a decomposition of multi-digit multiplication into necessary sub-tasks, and that this decomposition is what allows the Transformer to learn multi-digit multiplication.
>
> Thus, we re-ran all experiments (Figure 2, 3, 5, 6) on the “1-to-N” model: see Figures 2, 9, 11, 13. This in turn will introduce the following changes to our narrative:
>
> Rather than starting off with the premise of “Why can’t Transformers learn multiplication?”, we will discuss how different training data regimes (1-to-N, ICoT) allows the Transformer to learn 4x4-digit multiplication. By viewing both data regimes as a decomposition of the task into subtasks, we uncover mechanisms (i.e., attention tree, Figure 4) and feature geometry (i.e., Minkowski sums for partial products) that reflect these subtasks (i.e., pairwise products). Most importantly, we find that the correct long-range dependencies must be met in order to solve multi-digit multiplication – something that both the ICoT and 1-to-N models demonstrate, but the “standard training” model does not. With that in mind, we introduce an auxiliary loss model as one simple example to guide the model to learn long-range dependencies, which also allows the model to learn multi-digit multiplication.
>
> In summary, we have a total of 4 models being compared: one that fails at multiplication (SFT model), two that learn with the correct training data regime (ICoT, 1-to-N), and one that learns with an inductive bias via an auxiliary loss. We study *how* these models differ and find that the latter three successfully encode long-range dependencies, which is a critical component for multi-digit multiplication, and how task-decomposition in the training data can lead to different mechanisms and feature geometry.
>
> If you have further suggestions regarding the premise/narrative, please let us know. We are happy to continue the discussion!
>
> [1] Shen et al., "Positional Description Matters for Transformers Arithmetic," 2023.
>
> ### **Clarification on SFT**
>
> Thank you for this question - the term “SFT” can be confusing, especially in lieu of [1]. All of our models are 2 layer 4 head decoder-only models trained from scratch. There is no pre-training. The “SFT” model is just trained to predict the next tokens given training data in the form of 4x4 digit multiplications.
>
> We will change the term “SFT” to “Standard training” (ST) in our revision.

---

> > ### Author Response · Authors · 2025-11-21
> >
> > ### **Clarification on Presentation**
> >
> > Thank you for these points - your comments here are helpful, and we believe the revision has much better precision and clarity. Please see some of our additional comments below:
> >
> > Figure 3: Figure 3 presents the linear regression probe’s predictions for $\hat{c}$. In particular, at timesteps $2, …, 6$, we probe for whether we can predict $\hat{c}$ (as defined in Equation 2) from the model’s activations. The dashed red line represents the ground truth $\hat{c}$ values of the 1,000 different problems, while blue dots represent our probe’s predictions. The titles over each subplot specify the mean absolute error between the probe’s predictions and the groundtruth $\hat{c}$ values. Please let us know if anything is still unclear.
> >
> > Equation 4: Thank you for this feedback. We acknowledge that previously, our notations were confusing. We established more precise notations and rewrote Section 4.1 accordingly.
> >
> > In particular, we introduce notations to make clear what objects we are dealing with scalars ($\alpha$), vectors ($\mathbf{a}_x, \mathbf{b}_y \in \mathbb{R}^d$), and sets ($\mathcal{A}, \mathcal{B}$). We also define the needed operators between such objects (i.e., scalar multiplication on a set, sum between two sets, etc.).
> >
> > Please let us know if our revised writing is still confusing.
> >
> > Equation 5: We have added an entire appendix section to delve deeper into the relationship between the nested structure seen in Figure 5 and the covariance of $\mathbf{a}_x$ and $\mathbf{b}_y$. The $x$, $y$ indices are over digits $\{0, …, 9\}$ that $a$ and $b$ can take on. The newly added appendix discusses in more detail why Sums for $\mathbf{a}$ and Sums for $\mathbf{b}$ share the same eigenvectors, which basically boils down to the fact that both $\mathbf{a}$ and $\mathbf{b}$ are being derived from the same exact set of weights ($W_O, W_V$, token embeddings $E[\cdot]$).
> >
> > Please let us know if there are still parts that are confusing, as we greatly appreciate the feedback.
> >
> > Fourier Transform: In the Fourier analysis, $k$ often denotes the Fourier frequency index. Consider the model’s final next token prediction, which can be expressed as $\mathbf{z} = E\mathbf{h}^{L} \in \mathbb{R}^10$, $E \in \mathbb{R}^{10 \times d}$ is the model’s token embeddings for the 10 possible digits 0, …, 9, $\mathbf{h}^L \in \mathbb{R}^d$ is the model’s final layer.
> >
> > Let us treat $\mathbf{z}(n)$ as a function of digit $n$. $\mathbf{h}(n)$ can be expanded in a discrete Fourier series with up to 10 different frequency components $e^{-2\pi i \frac{kn}{10}}$, with $k \in \{0, …, 9\}$. (Note that $k$ can only go up to 9 because when $k \ge 10, e^{-2\pi i \frac{(k + 10)n}{10}} = e^{-2\pi i \frac{kn}{10}}$.)
> >
> > In practice we show that $E\mathbf{h}^L$ can be mostly reconstructed using just 4 of the frequency indices, $k \in \{0, 1, 2, 5\}$ (see Appendix B), which correspond to 6-dimensional real Fourier basis  ($\Phi(n)$).
> >
> > $\mathbf{1}(n) \equiv 1$ just indicates a function that always returns 1, regardless of what the input $n$ is.
> > $\mathbf{p}(n) \equiv (-1)^n$ does not indicate a vector, but is a simple function that returns +1 when n is even and -1 when n is odd.
> > We use these notations to make clear that every entry of $\Phi(n)$ is a function of $n$.
> > $*$ just refers to scalar multiplication - which we removed in our revision.
> >
> > We will make these clarifications in our revision.
> >
> > ### **Concern about Lack of Explanation for Success vs. Failure**
> >
> > Regarding the premise of Transformers not being able to learn multi-digit multiplication: as discussed above, we will change our framing to discuss how different training data regimes (ICoT, 1-to-N), which can both be thought of as task-decomposition, can lead to the proper mechanisms (i.e., long-range dependencies) that “standard training” (only training on 4x4 digit data) does not have.
> >
> > Regarding *why* versus *how*: we will modify our title and text to make our claims more precise, to ensure that we are not making any overclaims. To your point, we don’t provide a full explanation for *why* standard training does not learn the correct long-range dependencies. Instead, we study **how these models differ** and **how task-decomposition in the training data can lead to different mechanisms and feature geometry**.

---

### Official Review · Reviewer_tMVK · 2025-11-03

**Soundness:** 2
**Presentation:** 3
**Contribution:** 2
**Rating:** 4
**Confidence:** 3

**Summary:**

This paper studies why certain models (standard fine-tuned) fail at successfully performing multi-digit multiplication. The authors gain insight into this by reverse engineering a model that *can* successfully perform multi-digit multiplication via implicit chain of thought (ICoT). They uncover that the successful ICoT model encodes long-range dependencies, while the standard fine-tuned (SFT) models seemingly do not. Given these insights, they develop an auxiliary loss that provides a helpful inductive bias for the SFT model to learn long range dependencies to perform multi-digit multiplication.

**Strengths:**

1. Develop an understanding that leads to an actionable intervention: the auxiliary loss that leads to the SFT model getting up to 99% after previously being unable to perform the multiplication task.
2. The paper is well-written and the visualizations are well-made.
3. The authors include code with their paper.

**Weaknesses:**

W1. The present study is only conducted on 4-digit multiplication. Additionally, it's unclear how the auxiliary loss would generalize to multiplication with more digits e.g. 5x5, 6x6, etc.

W2. It's unclear how the insights would generalize to other tasks.

**Questions:**

Q1. Related to weakness 1: Do you have general results on multiplying numbers with more than 4 digits?

Q2: Do you have any idea why the case with the auxiliary loss goes up to 99% but not to 100% like the ICoT model? How does the auxiliary loss model perform as the number of digits increases? Does it stay at 99% consistently as the length increases or are there any changes?

Q3: How sensitive/robust is the performance with the auxiliary loss as you vary hyperparameters and random seeds?

Q4: I'm not sure if this is known in the literature already, or if you have thought about this, but do you have any insight on why the SFT model fails at the task in the first place? My understanding is that you diagnose the failure being related to the difficulty of the middle digits, but is there any understanding as to why the transformer with an autoregressive loss isn't able to do this?

Note: I find it intriguing that you use methods that on their own are not always robust for mechanistic understanding of exact model behaviour (e.g. PCA, linear probes, logit attribution, attention patterns), etc. but it still gives you insight to something important the model is doing because you're able to take a model that was previously unable to solve the task and significantly boost its performance on the task. Perhaps I'm not familiar enough with the mech interp literature in this area, but I personally find this interesting and it raises some philosophical questions for me on what is ``important'' for mechanistic understanding.

---

> ### Author Response · Authors · 2025-11-21
>
> Thank you for your careful review and for *noting the insight we offered for what is ``important’’ for mechanistic understanding*.
>
> **General Clarification on the Role of Auxiliary-Loss Model**: Before touching on each comment and question below, we first wish to make a general comment. We have noticed that many of the criticisms and questions revolve around our auxiliary loss model. To clarify, we view our main contribution as explaining what mechanisms and feature geometry allow the ICoT model to solve multi-digit multiplication. In doing so, we find that the ICoT model encodes the correct long-range dependencies, while the standard fine-tuned model does not.
>
> The **main purpose of introducing the auxiliary loss is mainly to validate our understanding**, in that correctly encoding long-range dependencies is a critical recipe for the model to learn multi-digit multiplication. For broader applications (outside of multi-digit multiplication), there may be more generic ways of adding an inductive bias to learn long-range dependencies, but our experiments are mainly to demonstrate that it is possible.
>
> With that being said, we respond to your comments and questions below!
>
> **W1: Request for Analysis of 5x5 and 6x6 Digit Multiplication**: Our experiments suggest that with a 2 layer 4 head architecture, neither ICoT nor the auxiliary loss is sufficient to learn 5x5 or 6x6 digit multiplication. According to [1], this would require re-running the entire setup with a 24-layer model (instead of our 2-layer mechanistic testbed). This would be computationally expensive and is not necessary to validate the point we are making.
>
> However, we do not view the study of larger architectures as a critical component for our study, as we are interested in what mechanisms and feature geometry allow the ICoT model to solve multi-digit multiplication.
>
> Furthermore, as noted above, we view our experiments with the auxiliary loss as a validation of our understanding, not so much as a proposed application.
>
> **W2: Concern about Generalization to Other Tasks**:
>  We view our main contribution as demonstrating a “weakness” of a standard training recipe (Transformer + autoregressive loss + gradient descent) on tasks that require long-range dependencies by fully reverse-engineering a controlled setup (multi-digit multiplication task).
>
> In doing so, we provide a few generalizable takeaways:
> * *“Task decomposition” can make a difference in the mechanisms and representations learned by the model.* Namely, ICoT can be viewed as decomposing multi-digit multiplication into many subtasks.
> As discovered during our rebuttal with reviewer cmTc, another way to train Transformers on multi-digit multiplication is to decompose the 4x4 digit multiplication into “1-to-N digit multiplication” (i.e., give training data for 1x4 digit multiplication, 2x4, 3x4 but also 2x1, 2x2, 2x3, etc.). **We call this model a “1-to-N” model, and repeat all of our experiments on the ICoT model on this 1-to-N model (see Figures 2, 9, 11, 13)**. In doing so, we find that the 1-to-N model also learns the correct long-range dependencies in order to solve 4x4 digit multiplication.
> All of this is to say that ICoT, 1-to-N training, and even the auxiliary loss model can be viewed as decomposing the task for the model, and such task decomposition can make a big difference in what (and how) the model learns.
>
> *  *We identify a key limitation of a common recipe in training contemporary models (Transformers + auto-regressive loss + gradient descent).* Although it would require a careful design to tease apart which of these components are responsible for the limitation of learning long-range dependencies, we speculate that the Transformer architecture does play a role. In particular, there is some evidence that RNNs and state space models (SSM) are better at capturing long-range dependencies, as it requires to maintain and update the correct state in its bottlenecked hidden representations [2, 3]. With that being said, for tasks whose solutions require non-local dependencies similar to multi-digit multiplication, the standard training recipe may not be a reliable choice. Our findings suggest it is valuable to explore new training paradigms for these tasks, for example, objectives that operate over full sequences (such as diffusion-style training), which may better align the learning signal with the global structure of the underlying algorithm.
>
> [1] Deng, Yuntian, et al. "Implicit chain of thought reasoning via knowledge distillation." arXiv preprint arXiv:2311.01460 (2023).
>
> [2] Tay, Yi, et al. "Long range arena: A benchmark for efficient transformers." arXiv preprint arXiv:2011.04006 (2020).
>
> [3] Gu, Albert, Karan Goel, and Christopher Ré. "Efficiently modeling long sequences with structured state spaces." arXiv preprint arXiv:2111.00396 (2021).

---

> ### Author Response · Authors · 2025-11-21
>
> **Q1: Request for Analysis of 5x5 and 6x6 Digit Multiplication**: We ran some additional experiments on 5x5 and 6x6 digit multiplication. Our experiments suggest that a 2 layer 4 head model is unable to learn these tasks, whether it’s with ICoT or the auxiliary loss. According to [1], this would require re-running the entire setup with a 24-layer model (instead of our 2-layer mechanistic testbed). This would be computationally expensive and is not necessary to validate the point we are making.  In addition, as mentioned previously, we see the value of the auxiliary loss experiment to validate our understanding that long-range dependency is a critical component that the model must learn.
>
> [1] Deng, Yuntian, et al. "Implicit chain of thought reasoning via knowledge distillation." arXiv preprint arXiv:2311.01460 (2023).
>
> **Q2: Auxiliary-Loss Model Accuracy**: For the model trained with an auxiliary loss, the 1% of incorrect cases mispredicts the middle digits by one or two. In the case of longer digits, we are seeing that a 2 layer 4 head model is unable to learn multiplication with longer digits - we plan on investigating this further during the remaining rebuttal period.
>
> **Q3: Robustness of the Auxiliary Loss Model to Hyperparameters and Seeds**: There are a few hyperparameters that are important to get the auxiliary loss model to learn properly. The first is the $\lambda$ term in Equation 8, which determines how much to weigh the auxiliary loss. After sweeping through a few values, we have converged to using $\lambda = 0.1$.
>
> Another critical hyperparameter is the learning rate (1e-4) and number of epochs (10), just as in any other supervised experiment. In particular, because we use a fixed set of training data (8M samples), we want to make sure the learning rate meets the right balance of being stable while being able to reach convergence within the specified number of epochs.
>
> With that being said, our training runs across 3 different random seeds all reach an accuracy of 98% or higher.
>
> **Q4: Understanding the Cause of SFT Failure**: We think this is a good question where a theoretical contribution could be made. In particular, we speculate that one possible explanation could be derived by viewing the multi-digit multiplication problem as a graphical search problem, where nodes represent the digits of the two operands and the edges represent the pairs of digits being multiplied to derive partial products (akin to the “attention tree” of Figure 4). Under such a perspective, the Transformer must find a correct graph that satisfies multi-digit multiplication. In the standard fine-tuning regime, it seems as though the model finds a graphical structure that allows the model to solve for early digits (c0, c1, c6, c7) but is unable to “back step” to try other graphical structures.

---

### Official Review · Reviewer_CsfC · 2025-11-03

**Soundness:** 3
**Presentation:** 3
**Contribution:** 2
**Rating:** 4
**Confidence:** 3

**Summary:**

The paper studies Transformers’ ability to learn tasks with long-range dependencies, focusing on 4x4 multiplication.
It compares a standard fine-tuned model (SFT) with a model trained using implicit chain-of-thought (ICoT).
With a 2-layer, 4-head architecture, ICoT reaches 100% accuracy, whereas SFT remains below 1%.
The core of the paper is a comparison of the internal mechanisms learned by SFT vs. ICoT.
Using logit attribution and linear probing, the authors argue that ICoT captures the required long-range dependencies while SFT does not.
To explain how ICoT computes these dependencies, the paper introduces an attention tree, revealing a sparse, binary-tree-like routing pattern that supplies the necessary tokens to compute $c_2$.
The paper then studies the geometry of ICoT’s hidden representations via PCA.
It finds that intermediate representations cluster by $a_i$ and $b_j$, and that the final hidden states exhibits a pentagonal-prism structure.
Additionally, the authors observe that SFT appears to get stuck in a local minimum, based on gradient norms and per-$c_k$ losses over training.
Finally, they introduce an auxiliary loss to predict $\hat{c}_k$, which enables SFT without CoT to learn the task successfully.

**Strengths:**

**S1.**
The paper is well written, with sufficient methodological detail.

**S2.**
Claims are validated through multiple analyses (logit attribution, linear probes, attention tree visualization, PCA, gradient norms and losses), which together provide strong support.

**Weaknesses:**

**W1.**
The paper does not fully address the fundamental reason why Transformers fail to learn multiplication when trained with SFT.
The results convincingly indicate that SFT fails to capture long-range dependencies and that ICoT succeeds, but they do not explain why SFT fails to develop those dependencies in training (e.g., optimization landscape, inductive bias).
For example, in line 375, the paper states that “the model is stuck in a local optimum”, and does not clarify the underlying cause.
At the current stage, the work feels closer to “an analysis of differences between SFT and ICoT”, than to a full answer to the title “why can’t transformers learn multiplication”.

Aside from this limitation, the work is interesting and solid.

**Questions:**

**Q1.**
I’m slightly unsure how Figure 5 was produced.
My understanding is: (1) run many problems; (2) collect the output vectors of a specific first-layer attention head; (3) run PCA; (4) color points by the digit at $a_i$ or $b_j$.
Which timestep is used for extracting output vectors?
Also, what does the purple box indicate in Figure 5(b)?

**Q2.**
How does Section 4 relate to the main message that ICoT learns multiplication while SFT fails?

**Q3.**
The pentagonal prism in Figure 6 is interesting.
Does this structure consistently appear across different ICoT runs (e.g., different seeds/initializations)?
Also, if the task were posed in a different base (e.g., 11 (prime) or 30 (many divisors)), what geometry would PCA reveal? (the second question is just out of curiosity, so you don’t have to run extra experiments for this question.)

**Q4.**
In Section 6, the SFT model with the auxiliary loss solves 4x4 problems successfully.
Could you provide Figure 2/5/6 style visualizations for this model?
This would help assess whether its internal mechanism aligns with ICoT or differs in important ways.

---

> ### Author Response · Authors · 2025-11-21
>
> Thank you for your careful review and for finding our work both interesting and solid.
>
>  > W1. The paper does not fully address the fundamental reason why Transformers fail to learn multiplication when trained with SFT.
>
> Our central contribution is in providing an explanation (by looking at both mechanisms and feature geometry) for how the ICoT model can solve multi-digit multiplication. This reveals the critical mechanism (or lack thereof) of long-range dependencies, which the standard fine-tuned model lacks.
>
> With that being said, you have a valid point: we explain *how* but not quite *why*. We will change our title, as well as claims throughout the paper, to reflect these points more precisely. We will also add the question regarding why the model gets stuck in a local optimum to the list of open questions in the discussion.
>
> We speculate that a theoretical explanation could potentially be derived by viewing the multi-digit multiplication problem as a graphical search problem, where nodes represent the digits of the two operands and the edges represent the pairs of digits being multiplied to derive partial products (akin to the “attention tree” of Figure 4). Under such a perspective, the Transformer must find a correct graph that satisfies multi-digit multiplication. In the standard fine-tuning regime, it seems as though the model finds a graphical structure that allows the model to solve for early digits (c0, c1, c6, c7) and is unable to “back step” to try other graphical structures. We leave a full theoretical account for future work.
>
>
> **Q1. I’m slightly unsure how Figure 5 was produced.**
> Yes, your understanding is exactly correct: we collect the output vectors from a specific attention head, run PCA, and label them based on either $a_i$ or $b_j$. In the case of Figure 5, we take the timestep when prompted for c2 (i.e., the last timestep of $a_0a_1a_2a_3 * b_0b_1b_2b_3 %%%#### c_0c_1$), but similar patterns of clustering based on digits are consistently seen across different prompts. In particular, whenever an attention head attends to a pair of digits, $a_i, b_j$, we can expect to see clusters that can be color-coded by $a_i$ and $b_j$.
>
> The purple box indicates a set of points belonging to prompts where $b_1$ equals a certain number, which is then zoomed in and demonstrated in Figure 5 (d). In Figure 5 (d), that same set of points is now labeled by $a_1$. We will clarify these confusions in our revision.
>
>
> **Q2: How does Section 4 relate to the main message that ICoT learns multiplication while SFT fails?**
>  Arguably, the representations of a model that has correctly learned a given task should reflect core characteristics of the task. With that being said, in Section 4 we ask, given the ICoT model that has learned digit multiplication, how do its representations allow the model to solve multi-digit multiplication? In doing so, we find intuitive representations that are suitable for multiplication: pairwise products (Section 4.1) and final digit predictions using Fourier bases (Section 4.2), which are not found in the SFT model.
>
> **Q3.1: Structure Consistency across Different ICoT Runs**: Yes, the prism appears consistently across ICoT runs. We trained with two additional seeds and observed the prism structure in both cases. See Figure 12 of our revision for examples.
>
> **Q3.2: How Different Bases Affect the Geometry**: Regarding what kind of structure we might expect to see with a different base, we think this is a fun problem. For base 10, its prime factors, 2 and 5, likely show up in the first 3 principal components, as it’s a very compact way for the model to express the 10 digits (combining mod 2 and mod 5). Now, before considering base 30 let’s consider base 15, which has factors 3 and 5. We think a reasonable structure, then, is for the model to express numbers by combining mod 3 (i.e., a triangle) and mod 5 (a pentagon). One way to visualize this in 3 dimensions might be to think of a pentagon repeated across 3 layers or a triangle repeated across 5 layers. Now extending to base 30, we could imagine adding another dimension of mod 2 (which certainly becomes harder to visualize in 3D).
>
> If we consider a prime number like base 11, our best guess is that the model would simply form a single ring to encode the 11 digits.
>
> **Q4. Additional Internal Analysis of Auxiliary Loss Model**: Yes, we added the equivalent of Figures 2, 5, and 6 for the auxiliary loss model in our revision. In particular, Figure 2 has been modified to include results for the auxiliary loss model, Figure 9 is equivalent to Figure 3, Figure 10 is equivalent to Figure 5, and Figure 13 is equivalent to Figure 6. In general, long-range dependencies are encoded in the model. However, we didn’t observe the Fourier structure here, which suggests that the auxiliary-loss model may be using a different mechanism to encode those dependencies.

---

### Author Response · Authors · 2025-11-21

We want to thank every reviewer for their time and careful review. We appreciate that all reviewers recognized our thorough analysis and find our insights interesting.

We have integrated most of your feedback into the revision, and believe it has made the paper significantly stronger. While we address each reviewer’s comments in individual responses below, here we highlight large scale changes made in the revision, which we believe will address concerns shared across all reviewers.

In particular, numerous reviewers pointed out that our work demonstrates *how* Transformers implement multi-digit multiplication, but does not answer *why* Transformers fail to *learn* it under standard training. We acknowledge these points and will make changes to our title and some of our claims to be made more precise.

We will sharpen our narrative to be more precise about our claims. Importantly, we have added the following experiments to illustrate our findings.

We have added another model (dubbed “1-to-N” model) that is also able to learn 4x4-digit multiplication. Rather than training on examples of 4x4 digit multiplication, this model is trained on samples of (1-to-N)x(1-to-N) digit multiplication (e.g., 1x4 digit, 2x4, 3x4, but also 2x2, 2x3, etc.). With this tweak in training data distribution, the “1-to-N” model is also able to learn 4x4 digit multiplication.

**We have repeated every experiment on this model**, and find similar results as our ICoT model (see Figures 2, 9, 11, 13). Most importantly, we find that the 1-to-N model also encodes the necessary long-range dependencies, which is what allows it to successfully solve 4x4 digit multiplication.

One can characterize both ICoT, 1-to-N training, and even the auxiliary loss as a decomposition of multi-digit multiplication into subtasks, which seems to be a common thread across all models that successfully learn multi-digit multiplication.

With that being said, we will change our narrative to be less about why Transformers fail to learn multiplication, but how task-decomposition (ICoT, 1-to-N training, auxiliary-loss) can lead to the right mechanisms. Interestingly, each training regime all have a common thread of successfully encoding the right long-range dependencies, but do learn slightly different representations (feature geometry).

Otherwise, the other change in our revision is in making notations much more precise.

To summarize our contributions, we study **how task decomposition affects the mechanisms and feature geometry learned by a model**. Decomposing the task in the form of ICoT, 1-to-N training, or even with an auxiliary loss can lead the model to the correct solution, which a model normally fails to reach otherwise. The successful models all share a common mechanism of encoding the correct long-range dependencies, but all learn different feature geometries.

Thank you again for your time - we are happy to continue the discussion!

---

### Author Response · Authors · 2025-11-29

We want to thank the new Area Chair for taking additional time to read our reviews and responses in light of this unusual circumstance. We greatly appreciate that our reviewers recognized our thorough analysis (as demonstrated by reviewer’s onsG rating of 10), found our insights interesting, and gave genuinely helpful feedback. In response to this feedback, we have **greatly updated our work and narrative**. To highlight the changes we’ve made, we summarize the feedback and responses to our reviewers.

In our work, we reverse-engineer a model trained with a specific regime (implicit chain-of-thought, aka ICoT) to understand why Transformers fail to perform multi-digit multiplication. With that said, here is a brief summary of our discussion with each reviewer.

### **Reviewer CsfC**
Reviewer CsfC notes that “the work is interesting and solid.”, although we don’t think their score reflects this sentiment.
Their main concern seems to be around our choice of wording, in which some of our text asks “why” transformers fail to learn multiplication, while we demonstrate “how” transformers fail (or succeed) at multiplication (see our General Notes below). We have updated our text to be more precise regarding this point.

Apart from this concern, the reviewer had asked for clarifications and a few extra experiments (i.e., the equivalence of Figures 2, 5, and 6 for two of our extra models can now be found in Figures 2, 9, 10, 11, 12, 13, 16), which we believe we have sufficiently addressed.

### **Reviewer cmTc**
Reviewer cmTc points out that apart from our ICoT regime, there is a second alternative way of teaching Transformers multi-digit multiplication (which we refer to henceforth as 1-to-N digit multiplication). Thus Reviewer cmTc points out that our claim that Transformers can't learn multiplication is too strong of a claim. Upon learning about the "1-to-N multiplication" regime, we've updated our text to avoid any overstatements. Additionally, **we've re-run every experiment with the "1-to-N multiplication model" and found that all of our results still hold**.

Apart from this main concern, we have cleaned up our notations significantly and believe that we have addressed Reviewer cmTc’s main concerns. We trust that the reviewer would have updated their scores under normal circumstances.

### **Reviewer tMVK**
Reviewer tMVK questioned the generalizability of our insights, and most of their questions revolved around experiments regarding the auxiliary loss.

**RE: Generalizability**: Regarding the experiments with 5×5 and 6×6, running additional experiments as suggested by the reviewer would require re-running the entire setup with a 24-layer model (instead of our 2-layer mechanistic testbed). This would be computationally expensive and is not necessary to validate the conceptual point we are making.

Additionally, we provide a few generalizable takeaways:

1) “Task deconstruction” can make a big difference in what (and how) the model learns a task. We provide evidence of this by studying the mechanisms and representations learned by the model for multi-digit multiplication. Namely, both ICoT and 1-to-N training can be viewed as deconstructing multi-digit multiplication into many subtasks. In our setting, we find that the model learns all the needed sub-tasks (i.e., partial products, Figures 4+5), and that these sub-tasks can be composed to encode the correct long-range dependencies (Figures 2, 3, 4).
All of this is to say, we demonstrate how task deconstruction can make a big difference in what (and how) the model learns.

2) We identify a key limitation of a common recipe in training contemporary models (Transformers + auto-regressive loss + gradient descent). Our findings suggest it is valuable to explore new training paradigms to tackle these challenges, such as better aligning the learning signal with the global structure of the underlying data.

**RE: Questions on auxiliary loss**: Regarding our auxiliary loss model, we wish to highlight that our main contribution is to demonstrate how Transformers perform (or fail to perform) multiplication, and to shed light on a limitation of Transformers. Long-range dependencies is a key necessity, and our experiments using the auxiliary loss is mainly to validate our understanding, which we repeat multiple times in our text.

### **General Notes**
Multiple reviewers noted that our title and text asks “why” Transformers fail to learn multiplication, but that our experiments demonstrate “how” transformers perform (or fail to perform) multiplication. We will update our text to be more precise regarding this subtle difference.

We believe that in response to our reviewers’ feedback, we have revised the work to be more rigorous and more precise. We would be happy to answer any questions and address any concerns you might have. Thank you again for your time!

---

### Meta-Review · Area_Chair_Qd1J · 2026-01-07

**Summary:**

This paper investigates how task decomposition enables Transformers to learn multi-digit multiplication through mechanistic analysis of models trained with implicit chain-of-thought and 1-to-N digit curricula. The authors reverse-engineer successful models to discover attention-based binary tree structures for caching partial products, Minkowski sum geometry with Fourier-based digit representations forming pentagonal prisms, and propose an auxiliary loss achieving 99% accuracy. Initial reviewer scores ranged dramatically from 2 to 10, reflecting disagreement about whether descriptive mechanistic analysis constitutes sufficient contribution. The comprehensive rebuttal addressed most technical concerns by adding 1-to-N model experiments, cleaning up mathematical notation, reframing the narrative from 'why transformers fail' to 'how task decomposition affects learning', and adding requested visualizations.

**Reviewer Concerns:**

Several concerns remain partially addressed or unresolved, and those include the theoretical explanation for why standard training fails to learn long-range dependencies remains incomplete despite gradient signals being present, generalization to longer digit multiplication, and the incomplete explanations for 99% vs 100% accuracy gap, etc. Generalization to other tasks beyond multiplication is discussed conceptually but not empirically validated.

**Reviewer Scores:**

While one reviewer awarded a perfect score, the remaining reviewers did not consider the work (or its findings) sufficiently novel. The negative reviewers don't seem convinced after the rebuttal. This large discrepancy likely reflects uncertainty about how broadly the work would be viewed as interesting in a conference setting. By directly addressing the reviewers' major concerns and extending the results beyond the single multiplication task, the paper could be positioned as more compelling and of wider interest.

---

### Decision · Program_Chairs · 2026-01-26

Reject